# Statistical analysis of hydrological response in urbanising catchments based on adaptive sampling using inter-amount times

Marie-claire ten Veldhuis[1,3] and Marc Schleiss[2,3]

[1]Delft University of Technology, Watermanagement Department
[2]Delft University of Technology, Geosciences and Remote Sensing Department
[3]Princeton University, Hydrometeorology Group

*Correspondence to:* Marie-claire ten Veldhuis (j.a.e.tenveldhuis@tudelft.nl)

**Abstract.** Urban catchments are typically characterised by a more flashy nature of the hydrological response compared to natural catchments. Predicting flow changes associated with urbanisation is not straightforward, as they are influenced by interactions between impervious cover, basin size, drainage connectivity and stormwater management infrastructure. In this study, we present an alternative approach to statistical analysis of hydrological response variability and basin flashiness, based on the distribution of inter-amount times. We analyse inter-amount time distributions of high-resolution streamflow time series for 17 (semi)urbanised basins in North Carolina, US, ranging from 13 km$^2$ to 238 km$^2$ in size. We show that in the inter-amount times framework, sampling frequency is tuned to the local variability of the flow pattern, resulting in a different representation and weighting of high and low flow periods. This leads to important differences in the way the quantiles, mean, coefficient of variation and skewness of the distributions vary across scales and results in lower mean intermittency and improved scaling. Moreover, we show that inter-amount times distributions can be used to detect regulation effects on flow patterns, identify critical sampling scales and characterise flashiness of hydrological response. The possibility to use both the classical approach and the inter-amount time framework to identify minimum observable scales and analyse flow data opens up interesting areas for future research.

## 1 Introduction

Hydrological response in urban catchments tends to be more flashy compared to natural ones as a result of their higher degree of imperviousness. Increase in flashiness is typically characterised by shorter response times to rainfall, higher runoff ratios and higher peak flows (Berne et al., 2004; Smith et al., 2005). On the other hand, high impervious degrees may reduce base flows and lead to intermittent flow during dry periods. At the same time, urbanisation is usually tied to development of urban drainage infrastructure, associated with artificial flow control as well as higher peak flows due to increased drainage connectivity. Predicting the degree of flashiness or base flow reduction associated with urbanisation is not straightforward, as it depends on the interplay of impervious cover, basin size and shape, soil properties, basin slope, drainage connectivity and control structures such as detention ponds, weirs and pumps (Emmanuel et al., 2012; Fletcher et al., 2013; Smith et al., 2013). Traditional analyses of flow time series tend to focus on specific aspects and flow characteristics, aiming for example at predicting low flow durations or peak flow magnitudes. For analysis of change in hydrological response, it may be beneficial to combine both

peak flow and low flow statistics into a single framework. This applies in particular to the context of urban hydrology where urbanisation and human intervention alter both high flow and low flow characteristics of the hydrological response. Combining both aspects in a single analysis is difficult, as flow distributions are highly skewed and frequencies of low and high flow values are very different. In this paper, we show how alternative sampling of flow time series based on inter-amount times leads to more balanced statistical distributions, better representation of both high and low flows in a single framework and more robust behaviour of statistical distributions across scales.

## 1.1 Statistical analysis of hydrological response

Many authors have investigated methods for characterising hydrological response and changes therein, including univariate analysis and multivariate statistics, combining several hydrograph properties such as flood peak, flood volume and flood duration (e.g., Salvadori and De Michele, 2004; Favre et al., 2004; Grimaldi and Serinaldi, 2006; Vittal et al., 2015). Traditional statistical analysis techniques tend to focus on either left or right tail properties of statistical distributions, but not necessarily using the same statistical framework. Low flow analyses for example are primarily concerned with the total time the flow stays below a critical threshold (see e.g. Smakhtin (2001) for an extensive review). By contrast, peak flow analysis puts more weight on total accumulated flows at a given time scale using annual flow maxima or peak-over-threshold values to derive extreme value statistics and establish flood frequency curves (e.g., Stedinger, 1983; Lang et al., 1999; Villarini et al., 2009; Smith and Smith, 2015). Both approaches are valid and solidly rooted in the context of extreme event analysis with numerous applications in drought and flood risk analysis. However, the statistical frameworks they rely on are not necessarily the same. Low flow analysis favours 'time' as a random variable. Peak flow analysis on the other hand treats the 'flow amount' over a fixed time interval as the main random quantity. This might seem more intuitive to many but there is no strong compelling reason to prefer one approach over the other a priori. For example, one might as well adopt an alternative framework in which the unknown random variable is the 'time' necessary to cumulate a fixed, critical amount of flow. This approach is known as the inter-amount time (IAT) method (Schleiss and Smith, 2016) and has been previously proposed to analyse the properties of intermittent rainfall time series. One of the main goals of this paper is to apply the IAT formalism to flow time series to derive properties of statistical distributions and compare the results to the ones obtained using the classical fixed-time framework.

## 1.2 Change in hydrological response, basin flashiness

An important characteristic that has been used to analyse change in hydrological response is basin flashiness, qualitatively described by Poff (2002) as one of the indicators characterising change in natural flow regimes and how this affects the ecological integrity of river ecosystems. Richter (1996) developed a set of 33 indices, the Indicators of Hydrological Alteration (IHA), including indicators for conditions associated with flashiness, such as frequency and duration of high and low pulses, and rate and frequency of change in flow conditions. Smith and Smith (2015) quantified flashiness of 5436 catchments in the contiguous United States based on peak flows exceeding $1 \text{ m}^3\text{s}^{-1}\text{km}^{-2}$ normalised flows (i.e., flows normalised by basin area). A frequently used index in the literature is the Richards-Baker (R-B) Flashiness Index (Baker et al., 2004), based on the

Richards pathlength (Gustafson et al., 2004). The R-B index is defined as the sum of absolute values of changes in flow values divided by the total cumulative flow, and is usually computed at the daily time scale. Similar to the coefficient of variation, it measures the relative dispersion of the flow at a given scale. A downside of the R-B index is that it highly sensitive to the scale of analysis. Baker et al. (2004) argued that for smaller basins ($< 50$ km$^2$) the use of hourly instead of daily flow data should be considered to compute R-B flashiness index, but also found that R-B flashiness values computed at hourly scale are highly sensitive to diurnal or other sub-daily low flow fluctuations. An important still unanswered question remains how to overcome scale sensitivity of flashiness indicators in different hydrological basins. This is crucial for establishing how urbanisation impacts flashiness and how changes relate to basin characteristics such as size, slope, imperviousness degree and whether urbanisation thresholds can be identified above which basin response is characteristically urban (Praskievicz and Chang, 2009).

### 1.3  Scaling analysis of hydrological flows

Scaling behaviour of river flows has been investigated by various authors, aiming to identify characteristics length and time scales and to detect scale dependence of hydrological response processes. Among the various statistical methods that have been proposed to investigate scaling, fractals and multifractals are among the most popular and powerful. Approaches for fractal analysis include: spectral analysis based on 2$^{nd}$ order properties, and trace moments analysis based on a wider range of statistical moments, typically between 0.1 and 4. The universal multifractal framework is based on the identification of scaling exponents summarising the changes in flow distributions across a given range of scales, (see Schertzer and Lovejoy (1987) and Schertzer and Lovejoy (2011) for a review).

One important drawback of multifractal analyses is that scaling of hydrological flow time series only holds in approximation and only over a limited range of scales. Many studies report the existence of "scale breaks" at which scaling parameters change and significant departures from (multi)fractality can be observed. Table 1 summarises findings from selected scaling analyses of flow time series in the literature. It shows that the number and location of the scale breaks as well as the values of the multifractal parameters are sensitive to the method applied to estimate them and the resolution of the data used to conduct the analysis. For example, Labat et al. (2013) performed spectral analysis and trace moments analysis for 30-minute flow times series and identified different flow regimes with scale breaks at 1 day for spectral and 16 hours for trace moments analysis. But when they performed the same analysis at daily and at 3 minute resolution, they identified different scaling regimes, with scale breaks at 16 days and 1 hour for daily and 3 minute resolution. Similarly, Sauquet et al. (2008) found different scaling regimes in their scaling analysis of flows for 34 basins, with scale breaks at 12 days for daily resolution and scale breaks varying between 8.7 hours and 7 days across basins when using hourly data resolution, based on spectral analysis. When they applied trace moments analysis for the same time series at hourly resolution, they found no scale breaks for the lower order moments and scale breaks between 10 and 150 hours for higher order moments. This shows that while most flows exhibit some sort of scaling behaviour, the identified scaling laws are not very robust nor consistent, as they are dependent on analysis methods and data resolution.

### 1.4 Statistical analysis of hydrological response based on adaptive sampling using inter-amount times

In this paper, the IAT formalism is applied to flow time series and statistical distributions and scaling properties are compared to the ones obtained using the classical fixed-time framework. To do this, we use flow observations collected in 17 hydrological basins in Charlotte, North Carolina. We aim to investigate what effects an adaptive sampling strategy such as IAT sampling has on statistical properties of the time series, in particular on the tails of the statistical distributions associated with peak flow and low flow extremes. The main problem with a fixed sampling rate, as in traditional flow time series analysis, is that it can only accurately represent frequencies of variations at time scales larger than a certain threshold. When frequencies higher than that exist, errors are introduced as information about the higher frequency variability is lost (Dippe and Wold, 1985). Increasing the sampling resolutions solves this problem, but results in oversampling of base flow values with respect to peak flows. An alternative consists in adopting an adaptive sampling strategy, i.e., one that adapts the sampling rate to the variability of the signal itself (e.g., Feizi et al., 2011). This makes sense for processes that are very unevenly distributed in time (such as rainfall and hydrological flows), and means taking more samples during periods of high activity (e.g., peak flows following storm events) and fewer during lower activity (e.g., periods of base flow). A well designed adaptive sampling technique lowers the probability of missing an interesting feature like peak flow and avoids oversampling during periods of small flow variations. We examine to what extent IATs influence the variance, skewness and shape of the sample distributions and how they can be used to better characterise basin flashiness and derive more robust scaling laws. Our results show that because IATs give more weight to rare peak flows compared to common base flows, they can provide different insights into flow properties and complement traditional flow time series analyses and metrics. Advantages of IATs sampling compared to conventional time series analysis are that IAT time series contain more information about peak flows and evolve in a more predictable way across ranges of smaller to larger scales. This makes them a more robust and reliable source of information to make predictions about flow characteristics at small, unobserved scales, including crucial information about rapidly evolving peak flows.

This paper is organised as follows, in section 2 we present the flow datasets and methods used for analysis. We explain the methodology for deriving normalised IATs and introduce metrics we used to compare properties of flows and IATs time series, to characterise hydrological response and compare response across basins. In section 3, results of the analyses are presented and discussed, first based on results obtained using a daily sampling scale, followed by results obtained a range of sampling scales, from hourly up to seasonal sampling scale. Conclusions and suggestions for future work are summarised in section 4.

## 2 Data and Methods

### 2.1 Flow datasets

The data used in the study were collected at 17 USGS stream gauging stations in Charlotte-Mecklenburg county, North Carolina. Gauging stations are located at the outlet of hydrological basins that range from 13 km$^2$ to 238 km$^2$ in size. The area is largely covered by low to high intensity urban development, covering 60% to 100% of basin areas. Percentage impervious cover varies from 8% in the least developed to 48% in the most urbanised basin covering the city centre of Charlotte. Figure 1

shows a map with the location of the area, boundaries of hydrological basins and location of stream gauges used in the analysis, table 2 summarises the main characteristics of the 17 basins.

Stream gage data were collected at 5 to 15 minute intervals over the period 1986-2011. Table 2 summarises the characteristics of the basins associated with each basin as well as the time period covered by the data. The temporal scale of observations changed from 15 to 5 minutes between 2010 and 2014, at different times for each gauge; overall 20-30% of the total observation record was covered by 5 minute intervals. Gauges measure water depth using pressure transducers and flow is derived using stage-discharge curves. These curves were established based on manual flow measurements during site visits and curves were checked and recalibrated during site visits several times per year. The percentage of missing flow data was smaller than 5% for all gauges included in the analysis; missing data were treated like zeros. The effect of missing data on IATs is difficult to predict as this depends on the pattern of missing values and whether or not they occur during a period of low or peak flow. Sensitivity studies by Schleiss and Smith (2016) have shown that the general effect of replacing missing values by zeros is that a few sample IATs will be overestimated. This mostly affects the right tail of the distribution and tends to have limited impact on peak flow characteristics. Another strategy would be to replace missing values by mean or median flow value, which may slightly reduce the overestimation of IATs in case several missing values occur in row. However, in this paper only the worst case scenario will be considered, i.e. missing values were replaced by zeros.

## 2.2 Definition of inter-amount times

In this paper we analyse hydrological flow variability, based on the distribution of inter-amount times. We use the following definition of inter-amount time (IATs), based on Schleiss and Smith (2016): Let $\Delta q > 0$ denote a fixed flow amount. We define the series of IATs $\tau_n(\Delta q)$ with respect to $\Delta q$ as follows:

$$\tau_n(\Delta q) = t_n(\Delta q) - t_{n-1}(\Delta q) \tag{1}$$

where $t_n(\Delta q)$ denotes the time at which the cumulative flow amount first exceeded $n$ times $(\Delta q)$:

$$t_n(\Delta q) = \inf\{u : Q(u) \geq n \cdot \Delta q\} \tag{2}$$

where $Q(u)$ denotes the cumulated flow at time $u$ and $Q(0) = 0$.

A steady flow pattern with constant flow has equal IATs for all values of $\Delta q$. A variable flow pattern, on the other hand, is characterized by a more variable IAT distribution.

## 2.3 Normalized inter-amounts

Flow magnitudes strongly vary from one gauge to another. To overcome this scale dependence and compare flow IATs across basins with different sizes and flow amounts, one needs to normalize IATs with respect to a common timescale. A possible way to do this is to fix an average IAT $\bar{\tau}$ (e.g., 24 h) and determine the inter-amount $\Delta q_{\bar{\tau}}$ at this timescale:

$$\Delta q_{\bar{\tau}} = \bar{\tau} \frac{Q_N}{T} \tag{3}$$

where $Q_N$ denotes the total cumulative flow amount at the considered location and T is the length of the studied time period. In other words, instead of comparing IATs for a fixed accumulation, we choose the mean IAT $\bar{\tau}$ and compute $(\Delta q)_{\bar{\tau}}$ such that the series of IATs $\{\tau_n(\Delta q_{\bar{\tau}}) : n = 1, \ldots, N\}$ has mean $\bar{\tau}$. Two locations with different cumulative flow amounts over a given period of time, e.g. over a year, therefore have different normalized inter-amounts.

5 ## 2.4 Sample estimates and minimum inter-amount scale

Inter-amount times can be estimated from a sample flow time series $q_1, .., q_N$ with temporal observation scale $\Delta t$ that may vary in time. But for simplicity, only the case with fixed temporal resolution $\Delta t$ will be considered below. A key step in this procedure is the determination of the first passage times $t_1, .., t_n$ in equation (2). This is done by considering the sample accumulated flow amounts $Q_1 < .. < Q_N$ at times $t_n = t_0 + n\Delta t$:

$$Q_n = \sum_{i=1}^{n} q_i \quad n = 1, \ldots, N \tag{4}$$

The exact first passage times $t_1, .., t_n$ for a fixed flow amount $\Delta q > 0$ are likely to be unknown due to the limited temporal resolution of the data. But we can approximate them based on linear interpolation:

$$\hat{t}_n(n\Delta q) = \Delta t \left( i_{n\Delta q} - \frac{Q_{i_{n\Delta q}} - n\Delta q}{q_{i_{n\Delta q}}} \right) \quad n = 1, \ldots, N \tag{5}$$

where $\hat{t}_n$ are the estimated passage times and $i_{n\Delta q}$ denotes the index (in the sample) at which the total cumulated flow first
exceeded $n$ times $(\Delta q)$:

$$i_{n\Delta q} = min\{i \in \mathbb{N} | Q_i \geq n\Delta q\} \quad n = 1, \ldots, N \tag{6}$$

The sample IAT estimates are then given by:

$$\hat{\tau}_n(\Delta q) = \hat{t}(n\Delta q) - \hat{t}(n\Delta q - \Delta q) \tag{7}$$

Because of the linear interpolation in (5), each sample IAT estimate, regardless of its length and the scale of analysis, will be
affected by a small interpolation error $\varepsilon_n(\Delta q) < \Delta t$. This error is random and has little influence on key statistics as long as IATs remain much larger than $\Delta t$, as is usually the case for large enough values of $\Delta q$ and during periods of low to moderate flow. Most of the interpolation errors happen during peak flows, when large flow amounts are accumulated over small periods of time. It is therefore important, for any given gauge, to identify the values of $\Delta q$ above which reliable IAT estimates can be derived. To identify the range of scales over which IATs can be reliably estimated, we consider the worst case scenario in
which all interpolation errors are equal to +/- $\Delta t$. In this case, the maximum relative error affecting IAT estimates is given by:

$$\varepsilon_n(\Delta q) = \frac{\Delta t}{\hat{\tau}_n(\Delta q)} \tag{8}$$

The minimum value of $\Delta q$ for which IATs can be reliably estimated depends on how strictly we want to control the estimation errors in (8). In our analysis, we set the mean of absolute relative errors to be smaller than 50%. This is a rather conservative approach as the estimation errors in (8) represent the worst case scenario and actual errors are likely to be much smaller than that. This leads to the following rule for determination of minimum inter-amounts $\Delta q$ that can be used for analysis:

$$\Delta q_{min} = min\{\Delta q > 0 : \overline{\varepsilon}_{\Delta q} < 0.5\} \tag{9}$$

where $\overline{\varepsilon}_{\Delta q}$ represents the arithmetic mean of the maximum relative errors in 8.

In addition to the lower bound, we also impose an upper bound on the inter-amounts used in our analysis. This is necessary to ensure IAT time series are long enough to compute relevant statistical moments. Typically, there should be at least 100 consecutive IATs, which yields the following upper bound for inter-amount $\Delta q$:

$$\Delta q_{max} = \lfloor \frac{Q_N}{100} \rfloor \tag{10}$$

where: $\lfloor \rfloor$ denotes the lower integer part and $Q_N$ is the total cumulative flow for the considered time series.

It is worth pointing out that the lower bound on the inter-amount in (9) also provides a rough indication of the left-tail properties of IATs, thus of the degree of flashiness of the hydrological response (i.e., the smallest scale at which flow variations can be studied given a fixed temporal observational resolution). More generally, the left tail properties of IAT distributions provide a good indication of what observational resolution is necessary to adequately capture the most extreme flow variations.
For more details on this important point, the reader is referred to the results section.

Note also that analyses of IATs were conducted for all gauges over the entire period of available data, without distinguishing between year, season or hour of the day. This was necessary as time series would otherwise be too short to study IATs across different scales. This means we mostly focus on average characteristics of IAT and flow distributions with respect to area
size and imperviousness degree and potential influence of flow regulation and stormwater detention facilities, as far as this information is available for the 17 basins. We refrain from investigating long-term trends, as our time series are restricted to maximum 30 years and because a recent study by Villarini (2016) showed no signs of long-term trends at 7506 gauges in the contiguous US in the last 30 years. Indeed, our own analyses revealed no significant long-term trend in mean IAT or flow variability over the considered time period.

## 2.5 Distribution of inter-amount times versus flows

Sample histograms of IATs and flows were analysed to investigate what different insights they provide into characteristics of the flow regimes. We plotted sample histograms for all gauges; appropriate bin widths were determined based on Scott's rule (Scott, 1979). We computed the coefficient of variation (CV), defined as the standard deviation divided by the mean, as an indicator for relative spread around the mean. Values of skewness and medcouple (Brys et al., 2004), a more robust skewness
metric based on ordered statistics instead of statistical moments, were computed to investigate asymmetry of the histograms and influence of outliers. We compared coefficient of variation, skewness and medcouple values for IATs with those for traditional flow time series and investigated relationships of the three statistics with basin area and imperviousness degree.

## 2.6 Distribution of changes in inter-amount times

First-order differences of IATs and flows were computed to look into characteristics of the rising and falling limbs of hydrographs. Because IATs are measured on an inverted scale, positive differences are associated with the falling limb of the hydrograph and negative differences with the rising limb of the hydrograph. Narrow ranges of histogram values for IAT differences indicate slowly varying flow; wide range histograms indicate more flashy behaviour. Positively skewed histograms for IAT differences indicate that the distribution is dominated by values on the rising limb and short recession limbs, while negatively skewed histograms indicate a larger part of the flow is associated with flow recession, i.e. long, slowly receding hydrographs, for instance induced by a strong groundwater flow component. Differences were computed at the 24-hour time scale, imposed by the minimum inter-amount scale rule. Similarly to the other histograms, bin widths was chosen based on Scott's rule.

## 2.7 Flashiness indicator and minimum observable scale

As mentioned earlier, the lower bound on the inter-amount provides a rough indication of left-tail properties of IAT distributions (i.e., short waiting times) and can therefore be used to characterise the degree of flashiness of the hydrological response. In flashier catchments, the flow can rise quicker, resulting in lower IATs during times of heavy rain. The minimum observable inter-amount represents the smallest scale at which flow variations can be studied with acceptable interpolation errors, given a fixed temporal observational resolution. By extension, the lower tail of the IAT distribution provides a good indication of what observational resolution is necessary to adequately capture the most extreme flow variations. The IAT flashiness indicator used in this paper is defined as the mean scale $\mu$ (expressed in hours) at which the 1% quantile of the IAT distribution equals the observational scale $\Delta t$ (15 min in our case). That is, the IAT flashiness indicates the average time needed to accumulate the amount of flow that can be accumulated in 15 minutes or less, 1% of the time. The larger the flashiness, the more flow can be accumulated over short amounts of time.

To better interpret results, we compared the IAT flashiness index with the classical R-B flashiness index defined in Baker et al. (2004):

$$R - B \; index = \frac{\sum\limits_{i=1}^{N} |q_i - q_{i-1}|}{\sum\limits_{i=1}^{N} q_i} \tag{11}$$

where: $q_i$ denotes the flow at time step $i$. The R-B flashiness index is dimensionless and can vary between 0 and 2. It is 0 for constant flow and 2 for highly variable and continuously changing flow. Its value is independent of the units chosen to represent flow (Baker et al., 2004). However, index values do depend on the time-scale at which they are computed, as will be discussed later in the results section. In our analysis, we computed R-B flashiness indices on daily aggregated flow values.

## 2.8 Scaling of inter-amount times

Multi-fractal analysis techniques were applied to investigate the scaling behaviour of IAT time series across different inter-amount scales. Multi-fractal analyses are based on the assumption of generalised scale invariance, in which the statistical moments or order $q > 0$ of a stochastic process $X_\lambda$ at scale ratio $\lambda$ are related by a power law:

$$\langle X_\lambda^q \rangle = C(q)\lambda^{K(q)} \tag{12}$$

where $\langle X_\lambda^q \rangle$ denote the moments of order $q$ of $X$ measured at a scale ratio $\lambda$, $C(q)$ is a constant (for each $q$) and $K(q)$ is called the moment scaling function. Within the universal multi-fractal framework, $K(q)$ is characterised with the help of only three parameters, $\alpha$, $C_1$ and $H$ (Schertzer and Lovejoy, 1987, 2011):

$$K(q) = \begin{cases} \frac{C_1}{\alpha-1}(q^\alpha - q) - qH & \text{if } \alpha \neq 1 \\ C_1 q \ln(q) - qH & \text{if } \alpha = 1 \end{cases} \tag{13}$$

]The parameter $C_1$ is referred to as the intermittency and characterises the clustering of the time series at smaller and smaller scales. $C_1 = 0$ for a homogeneous field that fills the embedded space and approaches 1 for an extremely concentrated field. The parameter $\alpha$ is called the multi-fractality index ($0 < \alpha < 2$) and it controls how the moments change when going from one scale to another. Finally, $H = -K(1)$ is called the Hurst exponent. Note that in the case of IATs, the mean inter-amount time $\overline{\tau}$ and scaling ratio $\lambda$ are inversely proportional to each other (i.e., $\Delta q_{\overline{\tau}} \sim \lambda^{-1}$). So either of them can be used here as a measure

of scale. The only difference will be the value of the constant $C(q)$ and the sign of the exponent in (12).

The scaling quality is assessed by noting that if (12) is true, the log-moments for fixed values of $q$ should be a linear function of the log-scale:

$$\ln(\langle X_\lambda^q \rangle) = K(q)\ln(\lambda) + ln(C(q)) \tag{14}$$

The extent to which this equality holds can be assessed by fitting a linear regression model and computing the $R^2$ values,

i.e., the coefficient of determination of the log-moments versus the log-scale for each value of $q$. A $R^2$ of 1 indicates perfect scaling. The lower the coefficient of determination, the larger the deviations from scale-invariance. The approach was repeated for different values of $q$ and the mean or minimum value of $R^2$ were chosen as a way to assess the overall quality of the scaling. Based on recommendations by Lombardo et al. (2014), we refrained from using too low or high order moments and only considered values of $q$ between 0.4 and 2.5, with an equal number of moments above and below 1 to avoid favoring one

tail of the distribution over the other. The range of IAT scales that was used for the analysis was constrained by the length of the time series and the minimum and maximum inter-amounts defined in (9) and (10). The corresponding scales varied from 0.1 to 0.6 days up to 28 to 100 days for the longest time series.

## 3 Results

In the following sections we compare statistical properties of flow and IAT time series and highlight differences that result

from the different sampling strategies. Analyses are first conducted at the 24 hour time-scale and associated mean inter-amount

sampling scale. In the second part of this section, we analyse how statistical properties of flow and IAT time series vary across scales and quantify flashiness and scaling behaviour of both time series.

## 3.1 Time series and variability analysis of inter-amount times and flow values

Figure 2 shows an example of times series for flows and for IATs for the gauge at Taggart Creek, a 13.6 km$^2$ basin in the Charlotte catchment, at 24 hour sampling scale. The two graphs bring out different aspects of flow variability: flow time series have most of their data points concentrated in the low flow region, with intermittent peak flows characterising rain events. For IATs, peak flows appear as minima, while periods of low flow show up as maxima in the time series. The graph illustrates how IAT samples are more evenly distributed across high and low values in the time series compared to flows. The mean inter-amount for Taggart Creek at 24-hour sampling scale is 13,559 m$^3$, equivalent to 0.998 mm when normalised by basin area. Hence, in IAT analysis, the time series is sampled each time 0.998 mm of normalised flow has been accumulated, which amounts to frequent samples during high flows and fewer samples during low flow periods. For instance, a high concentration of IAT samples is clearly visible for the wet year 2003: this year is represented by 802 IAT samples compared to the 365 samples per year we have on average.

Figure 3 illustrates the adaptive sampling strategy based on flow amounts as the sampling unit, instead of fixed time steps. Figure 3b shows cumulative flow over a week, where a storm event occurred on 7 August. In conventional flow time series analysis, flow is sampled daily (in this example), resulting in one sample representing the peak period of the event (i.e., on 7 August). In IAT analysis, flow accumulation determines the sampling frequency, so periods of low flow are sparsely sampled, while the storm event is represented by eight samples. This illustrates how, even for 24-hour mean inter-amounts, sampling frequency can be much higher during periods of peak flow.

Histograms of flow time series and IATs at daily time scale are plotted in figure 4, for two basins, Taggart Creek (13.6 km$^2$) and LSugarA (111 km$^2$). The corresponding inter-amounts are 1 mm and 1.8 mm of normalised flow (for Taggart and LSugarA, respectively). Histograms for the other 15 basins are available in the supplement to this paper. Figure 4 shows that both histograms of flows and IATs are positively skewed. In both cases however, left and right tails represent very different flow characteristics. The left tail of the flow's histogram essentially features common base flow values while the right tail captures rare peak flow events. By contrast, the left tail of IAT distributions, which makes up most of the values, predominantly features short IAT values associated with periods of high flow. The rare samples that make up for the right tail represent long waiting times associated with extended periods of low flow. The low density of the first bin in the flow histogram for LSugarA reflects the effect of low flow regulation for this basin. The same effect is reflected in the bi-modal shape of the IATs histogram. Note that the low density 0-0.5 bin in the flow histogram for LSugarA corresponds to the >3.5 day bins in the IAT histogram.

Tables 2 (6$^{th}$ and 7$^{th}$ columns) and 3 summarise statistics of flow and IAT time series, at 24 hour sampling scale. The results show that mean inter-amounts vary from 12,275 m$^3$ for the smallest to 269,534 m$^3$ for the largest basin in size. Mean normalised inter-amounts vary from 0.68 mm for Irvins Creek, the least urbanised basin (8.2% imperviousness) to 1.79 mm for

Little Sugar Creek at Archdale, one of the largest basins with a high degree of imperviousness (32%). Coefficients of variation at the daily scale are consistently higher for flows than for IATs (e.g., 1.7 times higher on average), which highlights the more balanced nature of IAT distributions. Skewness values at the daily time scale are 3.6 times higher for flows than for IATs, on average, and even up to a factor of 15 higher for Stewart Creek. By contrast, medcouple values for flows are lower than for IATs by a factor 2.1 on average. This shows that statistical distributions of flows are strongly influenced by the presence of a few very large outliers. Most of the weight, however, lies close to the median (low medcouple). The IAT sampling gives more weight to rare peak flow values and less to common base flow, therefore producing distributions with lower skewness and more information about peak flow values. The larger medcouple values mean that IATs above the median value tend to be much further away from the median than values below the median. In other words, the right part of the distribution, which features long waiting times during low flow conditions can be very stretched.

These results show that adapative sampling based on inter-amounts leads to more balanced representation of high and low flows, resulting in lower coefficients of variation reflecting stabler statistical variance compared to traditional flow time series sampling. We like to point out that these results were obtained at the 24 h sampling scales. In section 3.4, behaviour of the statistical distributions of flows and IATs, as well as associated CV, skewness and medcouple values will be analysed across a range of subdaily to seasonal scales.

## 3.2   Statistical distribution properties comparison across different hydrological basins

Subsequently, we compared properties of IAT and flow distributions across the 17 basins in relation to basin characteristics. Figure 5 shows scatter plots of mean normalised inter-amounts, CV, skewness and medcouple values for flows and IATs as a function of basin area and imperviousness degree. The results show a positive correlation of 24-hour mean normalised flows or inter-amounts with basin size (Spearman correlation 0.55). This is mainly explained by a lower likelihood of low flows that have a large influence at this scale (24 hours). Mean normalised flows correlate positively with imperviousness degree (Spearman correlation 0.58), which is likely to be explained by a generally growing importance of flow regulation, resulting in maintenance of higher mean base flows in urbanised basins.

Looking at CV-values across all basins (figure 5. c, d), we found that CV-values for both flows and IATs generally decrease with basin size and with imperviousness degree. CV-values are significantly negatively correlated with basin size for flows (Spearman rank correlation -0.75). This can be explained by an increased smoothing effect on flow variation, in particular a lower likelihood of low flow extremes during dry periods for larger basins. CV-values for IAT distributions are significantly negatively correlated with imperviousness (Spearman rank correlation -0.57). Since IAT distributions put more weight on high flows compared to low flows as a result of their adaptive sampling strategy, this probably indicates stronger influence of flow regulation in urbanised basins resulting in more uniform runoff during rainy periods. IATs during these periods concentrate relatively more closely to the mean and show fewer extremes (this is clearly visible for the most urbanised basin, LSugarM, gauge 409). The effect of urbanisation as reflected by imperviousness degree on IAT statistics appears to be more important than basin size.

Scatter plots for skewness and medcouple values (figure 5. e, f, g, h) show generally weak correlation with basin area (Spearman correlations not significant at the 5% level). Skewness of IAT distributions is significantly negatively correlated with imperviousness (Spearman rank correlation -0.63). Similar to CV-values, this probably indicates stronger influence of flow regulation on flows in urbanised basins. Medcouple values for IATs clearly show three low value outliers: for Stewart Creek (970), LSugarP (530) and LSugarA (507). In these basins, active low flow control is applied[1] preventing occurrence of low flow extremes and high IAT extremes. The effect shows up more clearly for IAT medcouple values, as a result of the adaptive sampling strategy that gives more weight to peak flows, leading to generally higher medcouple values, but also reflecting more clearly the absence of low flow extremes. Some of the basins in this study are subbasins of each other, which implies that flows can be correlated. Table 4 summarises CV, skewness and medcouple values for three sets of subbasins in the Charlotte catchment. The results show that variability in skewness and medcouple values is unrelated to inter-basin connections. The same applies for flow CV-values, while CV-values for IATs seem to be clustered by group of subbasins, indicating that inter-basin correlation plays a role in explaining IAT $2^{nd}$ order variability. The fact that the effect is only visible for IAT, not for flows, indicates that correlation is mainly associated with occurrence of peak flows, that receive more weight in IAT than in flow statistics

In this section we discussed distributions of IATs and flows at the 24 hour scale. Results showed that larger basins are generally characterised by stronger smoothing of flows, resulting in higher mean flow, lower CV and lower skewness of the flow histograms. Flow variability is clearly correlated with basin size, which is mainly a result of smoothing of low flows, in the left tail of the flow histogram. Results showed that larger imperviousness is associated with higher mean flows and significantly lower CV-values for IATs, which is mainly associated with stronger flow regulation by dams and detention ponds in urbanised basins. CV and skewness values are much higher for flows than for IATs, while medcouple values are lower for flows, indicating strong asymmetry of the flow distributions and low representation of high flow extremes in the statistical distribution.

### 3.3 Distribution of changes in inter-amount times

Figure 6 shows histograms of first-order differences in IATs and flows at the 24 hour analysis scale, for Irvins Creek, the least urbanised basin, LSugarM the most impervious basin, Stewart Creek, a basin with low flow regulation and McAlpine, the largest of all studied basins. In the flow histograms, negative differences are associated with recession, positive differences with flow rise. Conversely, negative differences in IATs occur during flow rise, positive differences during flow recession. Most flow differences are concentrated in the 0 to -0.5 mm bin, associated with slow flow recession of 0.5 mm/day. Most IAT differences are concentrated in the 0 to 0.1 or 0.2 day bin, associated with steeper flow recession of approximately 5 to 10 mm per day. This reflects the relatively higher sampling of rapid flow response for IATs compared to conventional flow sampling. Skewness and medcouple values of the histograms provide indications of hydrograph shape, in particular of the steepness of the hydrograph recession limb: higher skewness, thus more weight of the distribution concentrated in one of the tails, indicates slow flow

---

[1]USGS, water year reports

recession compared to relatively rapid flow rise. Figure 7 shows scatter plots for skewness and medcouple values versus basin size and imperviousness, for all basins. The three basins with low flow regulation (970, 530, 507) can be recognised by their low medcouple values for IAT difference indicating near symmetrical histograms, i.e. flow rise and recession occur at similar rates. Most IAT differences histograms are negatively skewed, with a longer left tail than right tail, i.e. IATs generally decrease

quicker (flow rise) than they increase (flow recession). Strongest negative skewness for IAT differences was found for the least urbanised basin (Irvins Creek, gauge 975), indicative of steep flow rise occurring in this basin. Significant positive correlation was found between skewness of IAT difference histograms and imperviousness (Spearman correlations 0.75), indicating lower probably of steep flow rise in higher urbanised basin. Negative correlation was found between medcouple and imperviousness (Spearman correlation -0.55), thus relatively more symmetrical hydrographs with flow rise and recession at similar rates occur

for urbanised basins. Here, subbasin correlation appears to play role: medcouple values are higher overall in the McAlpine subbasins than in Little Sugar Creek and Irwin subbasins (see table 4). Significant correlations of IAT differences skewness and medcouple with imperviousness show that urbanisation is associated with more regulated flows, confirming findings in section 3.1.

## 3.4  Inter-amount times variability across scales, from sub-daily to seasonal sampling scale

In this section we analyse the variability of IATs and flows across a wide range of sampling scales. We investigate how the statistical distributions and hydrological response characteristics change when moving from inter-event (multiple days) to intra-event (sub-daily) scales. Figure 8 shows quantile plots for normalised flows and IATs at scales between 12 hours and 64 days, for Taggart Creek. On the horizontal axis is the sampling scale, i.e. fixed sampling time for conventional flow statistics or, equivalently, mean inter-amounts for IAT statistics. Note that for the IAT analysis, mean inter-amounts are normalised by

basin area size and reported in mm to allow easier interpretation of flow magnitudes and to allow easier comparison between basins. For instance, the normalised inter-amount $\Delta q$ for Taggart Creek at the daily scale is 0.998 mm. The vertical axis shows quantiles of normalised flows respectively IATs corresponding to the sampling scale in time or $\Delta q$. Values on both x- and y-axes are plotted on log scales to allow easier visualisation of quantile values that vary by 2 to 4 orders of magnitude. The bold black line denotes the mean, dotted black line shows median values. The central part of the quantile plots represents the

25-75 percentile range, upper and lower whiskers 10-90 percentiles and crosses the 1 and 99 percentiles.

We can see that mean values of normalised flows and IATs decrease log-linearly with sampling scale, as indicated by a straight line in the log-log plot, i.e. the sampling mean follows power-law scaling. As histogram analysis at the 24 hour scale already showed, statistical distributions of both flows and IATs are highly skewed. Moreover, skewness increases at smaller scales as indicated by an increasing distance between mean and median values. Median values for flows follow close

to log-linear scaling (albeit steeper compared to the mean) but exhibit stronger departures from log-linear scaling for IATs. In particular, the median of IATs shifts from close to log-linear scaling between 16-64 mm (associated with about 16 to 64 days) to non-log-linear scaling between 1-14 mm scales (1-14 days) and again to near-log-linear scaling below 1 mm. Coincidentally, these transitions correspond to the range of scales over which IATs generally transition from being inter-event to intra-event dominated. Indeed, IATs at coarser scales mostly combine the properties of multiple storms, resulting in a more symmetric

distribution. This effect is much stronger in IAT than in flow distributions, because it is mainly associated with changes in sampling of peak flows which are more frequently sampled in the IAT framework than in the conventional fixed time approach.

Comparing the 10-90 and 1-99 percentile ranges in figures 8a and 8b we see that the 10-90 percentile range of IATs gradually increases towards smaller scales. For flows, the 10-90 percentile range remains approximately constant, however, distance

between 90 and 99 percentile values rapidly increases towards smaller scales. This reflects the highly skewed nature of flow distributions caused by oversampling of low flows compared to high flows; an effect that increases progressively towards smaller scales. By contrast, 10-90 and 1-99 percentile ranges for IATs increase more or less similarly with scale, for sampling scales ranging from 0.51 mm to approximately 10-16 mm. This indicates that the tails of IAT distributions are more or less equally sampled, at least up to the 1 and 99 percentiles.

The upper 75, 90 and 99 IAT percentiles of IATs, associated with low flow periods, change approximately log-linearly with scale, showing that upper tail percentiles of IAT values refer to the same low flow periods across all scales, up to 8-16 mm scale. Associated low flows are approximately 0.1 mm/day. The 1-percentiles for flows are associated with approximately 0.02 mm/day, for the 12 hour to 4 day scale, showing that the distribution tail associated with low flows captures lower flow extremes in conventional sampling than in IAT sampling. This is a result of the relatively high frqeuency at which low flows are sampled.

Conversely, peak flows, associated with the right tail of the flow distribution are sampled less frequently in conventional flow sampling: the 99 percentiles are associated with peak flows of 0.78 to 0.38 mm/h for 12 hour to 4 days scale. The 1-percentiles of IATs are associated with peak flows of about 20 mm/h, at the 0.5 to 4 mm inter-amount scale, associated with mean IATs of 12 hours to 4 days. This shows that the IAT distribution captures more extreme peak flow values than conventional flow sampling, at the same sampling scale.

Quantile plots of inter-amounts over range of scales were created for all 17 gauges included in our analysis (results are added as a supplement to this paper). This allowed us to compare transition ranges between inter-event dominated and intra-event dominated IAT distributions for all basins. Results show that for 10% IAT quantiles, the lower end of the transition range, where intra-event characteristics start to be mixed with inter-event phenomena, lies roughly between 10 mm and 25 mm mean

inter-amounts, being accumulated in about 1 hour in most of the basins. Lower values are found for basins with higher urbanisation degree and for basins where low flow control is applied, reflecting the smoothing influence of flow control measures on peak flows. Similarly, one can compare the amount of flow that is being generated in an hour, compared to the mean flow. This can be derived from the IAT quantile plots by looking at the scale at which a given IAT quantile, for instance 10% or 1%, equals 1 hour. For Taggart Creek, the IAT 1-percentile equals 1 hour at sampling scale of 18 mm of mean normalised flow or

equivalently, 18 days of mean IAT. This means there is a 1% probability of exceeding 18mm of flow accumulation in 1 hour or less. Or, in terms of time it implies that there is a 1% chance to accumulate the amount of flow measured on average over a period of 18 days in 1 hour or less. Thus, higher values of 1 hour, 1-percentiles indicate stronger flashiness of basin response. Comparing values across basins, we found that higher values of 1%, 1 hour accumulations were strongly correlated with basin area, while no significant correlation with imperviousness was observed.

Subsequently, we investigated scaling behaviour from the perspective of statistical moments, by looking at coefficients of variation for flows and IATs across scales. For the purpose of statistical analysis and downscaling applications, it is important to have a robust scaling model, that predicts how distributions change when going from one scale to another. Scale invariance means that a distribution can be derived at any scale, especially small scales, by shifting and scaling the distributon at larger scales. One way to assess the property of scale invariance is to check if the statistical moments of distributions follow a power-law of scale. Figure 9 shows coefficients of variation, computed as the ratio of the $2^{nd}$ over the $1^{st}$ order moment, for four gauges, across a range of sub-daily (3 to 12 hours) up to bi-monthly (60-68 days) scales. Results show that coefficients of variation for flows vary non-linearly with scale, while they approximately follow a power law with scale for IATs. For Irvins Creek, the most natural basin in this study (8.2% imperviousness, figure 9a), CV-values of IATs and flows are similar over a range of 10 to 50 days. At smaller scales, CV-values for flows increase more rapidly than for IATs, indicating that IAT variance remains more stable at smaller scales, while variance rapidly increases at small scales for flows, as a result of growing skewness of the statistical distribution, caused by relative oversampling of low flows, or conversely, undersampling of high flows. CV-values for Upper LSugar Creek, the most urbanised basin are lower than for Irvins Creek, especially at smaller scales (figure 9b). This is explained by the influence of flow control measures in this basin, as flows are constrained by the stormwater drainage system. The difference is more pronounced for IATs, because IAT variance is more sensitive to peak flows as a result of the adaptive sampling strategy. Figure 9c shows that for LMcAlpine, the largest basin (238.4 km$^2$), CV-values for flow are more or less stable between 3 and 24 hour scale, due to strong smoothing of peak flows at this intra-event scale. In contrast, CV-values for IATs increase over this range, due to scale sensitivity of the upper tail of the IAT distribution, where long IATs at this small scale (0.1 to 1.1 mm for 3 to 24 hours) are broken up more unevenly, creating increased CV and skewness. This shows that for analysis of low flows, especially in basins characterised by strongly smoothed flow variability, IAT analysis offers little advantage and conventional flow statistics are more suitable. CV-values for Stewart Creek in figure 9d show very low CV-values for IATs that vary little with scale, while CV-values for flows are much higher and strongly sensitive to scale. Stewart Creek is a small, semi-urbanised basin (33% imperviousness) where active low flow control is applied. This results in very low variability in IATs across the entire range of scales, while CV-values for flows are lower than those for similar basins, but highly sensitive to scale, probably due to unbalanced sampling of peak flows compared to very stable low flows.

In section 3.1 we analysed skewness and medcouple values of flow and IAT distributions at the 24-hour scale and found that skewness values were lower and medcouple higher for IATs than for flows. This was explained by the sensitivity of flow distributions to rare peak flows compared to frequently sampled low flows. Initial analyses of skewness and medcouple values across scales showed that results are highly sensitive to the sampling scale. While CV-values show a stable pattern across scales, results for skewness and medcouple are much more variable, across scales and across basins. Explanation of this scale sensitivity of skewness metrics and what information can derived from this about the tails of the distributions requires deeper analysis that will be part of future work.

## 3.5 Flashiness indicators and minimum observable scale

Two flashiness indicators were computed, as explained in section 2, the classical R-B flashiness index and an IAT flashiness indicator based on characteristics of the IAT distribution. Table 3 summarises flashiness values for all gauges, as well as minimum and maximum observable inter-amounts, as defined in equations (9) and (10). IAT flashiness indicators vary between 12.5 and 165 hours; higher values are generally associated with smaller basins. -B flashiness values vary between 0.8 and 1.3, indicative of moderately variable flows (R-B flashiness can vary between 0 and 2). Values are in the same range as those reported by Baker et al. (2004) for smaller basins: they found R-B flashiness values larger than 1 for basins smaller than 50 km$^2$). R-B flashiness is strongly correlated with CV-values (figure 10c, Spearman correlation 0.77); this confirms that R-B flashiness is essentially a metric of flow variability. Figure 10a shows that IAT-based flashiness and R-B flashiness are moderately correlated (Spearman rank correlation 0.55), yet there are some striking differences. The three low-flow-regulated basins have very low R-B flashiness values, while IAT flashiness values are in line with values for other basins. This is explained by R-B flashiness being strongly sensitive to low flow variability, while IAT flashiness is more sensitive to occurrence of peak flow values. For instance, McAlpine basin (gauge 255) has a very high IAT flashiness as a result of high occurrence of peak flows. On the other hand LSugarM (gauge 409), the most urbanised basin, has low IAT flashiness as a result of peak values being capped by maximum capacity of pipes in the drainage network.

Figure 10b and 10d shows scatter plots of IAT flashiness (left y-axis) and R-B flashiness (right y-axis) versus basin area and imperviousness, for all gauges. They show a clear relationship between flashiness and basin area (Spearman correlation -0.83 for IAT, -0.71 for R-B flashiness), with a large range of flashiness values for the smallest basins (< approx. 30 km$^2$). Here, clearly other processes than basin size play a role in explaining flashiness. Correlations between R-B and IAT flashiness versus impervious degree are not significant at the 5% level. For R-B flashiness, the most pervious and the most impervious basins (gauges 975 and 409 respectively) are both in the high range of flashiness values, showing that other influences, such as basin size and presence or absence of low flow regulation play a more important role than imperviousness degree. IAT flashiness tends to decrease for a combination of higher imperviousness and large basins, basin size playing a stronger role than urbanisation. The most urbanised basin, LSugarM (gauge 409, 31.7 km$^2$, 48% imperviousness) has a relatively low flashiness value of 48.8 hours, while the least impervious basin, Irvins Creek (gauge 975, 21.8 km$^2$, 8% imperviousness) has a high flashiness value of 102.8 hours. As discussed in section 3.1, the effect of urbanisation on flow patterns for the basins in the study area seems to be mainly determined by increased flow regulation associated with introduction of dams, stormwater detention basins and stormwater drains with capacity limitations. While higher imperviousness leads to higher mean runoff flows (for instance, 1.5 mm for LSugarM versus 0.68 mm for Irvins Creek, at 24 hour scale), rainfall in impervious basins tends to run off relatively more quickly and uniformly, depending on the degree of flow regulation. The leads to a mixed effect of basin size, imperviousness and flow regulation on IAT flashiness and peak flows.

In this study, IAT flashiness values were defined as the time that is needed on average to accumulate the amount of flow that is accumulated in 15 minutes or less, 1% of the time. R-B flashiness indices were computed at the daily scale, to allow

comparison with results obtained by Baker et al. (2004). For a fair comparison, both flashiness indices should be computed at similar scales, as far as possible, given that definitions used in the two approaches are different. We aimed to compute both indices at hourly scale, as this is an appropriate scale in relation to the size of most of the basins in our analysis and a reasonable compromise between the 15 minute and 24 hour time scales used for IAT flashiness and R-B flashiness index respectively. Note that Baker et al. (2004) stated that the hourly scale would be more suitable for smaller basins (<30 km$^2$), but never computed R-B flashiness values at this scale, only Richard's pathlengths. When we computed R-B flashiness indices at the hourly scale, using the same definition, we found lower flashiness than at the daily scale, which is rather counterintuitive, as one would expect higher flashiness at smaller scales due to the fact that Richard's pathlengths increase from daily to hourly scales. However, R-B flashiness is based on absolute differences of flow values, not gradients (i.e., differences per unit of time). And since flow differences decrease when moving toward smaller scales, R-B index also decreases. Alternatively, one could use discharges instead of flow amounts, but then values could grow much larger than 2. Regardless of the used approach, R-B flashiness index appears to be rather sensitive to the scale of analysis. By contrast, the IAT flashiness index proposed in this paper tends to be much more robust. Additional sensitivity analyses (not shown) revealed almost no changes in IAT flashiness estimates for 15 minutes to 3-6 hours aggregation scales. Beyond that, significant underestimation started to occur as the resolution is not sufficient anymore to correctly capture peak flow variability. For data aggregated at 24 h resolution (instead of the original 15 min), IAT flashiness values were underestimated by 20-80%, depending on the considered gauge.

Quantile plots of IAT distributions furthermore provide information about the minimum observable scale at a given observational resolution (15 minutes, in the data series used in our analysis), i.e. the degree of flow variability that occurs at scales smaller than the observation scale. When moving towards smaller sampling scales, a growing percentage of flow accumulations occurs in less than 15 minutes, hence cannot be analysed at the given observational resolution. This typically coincides with peak flows and implies that during peak events, the observational resolution is too low to measure flow variability. IAT analysis can thus be used to identify a critical resolution for flow observations, if a given peak flow accumulation is of interest. This could be associated with for instance the capacity of detention ponds or flooding caused by exceedance of stormwater drainage capacity. For the example of Taggart Creek (figure 8b), the scale at which 1% of flow accumulations occurs in less than 15 minutes is associated with inter-amount sampling scale of 4.76 mm. This implies that flows that exceed 4.76 mm in 15 minutes, i.e. peak flows above 19.0 mm/h, cannot be observed 1% of the time. If correct observation of peak flows of this magnitude or larger is important, flow data need to be collected at a higher than 15 minute resolution during times of peak flows. This is typically the case of urban basins, where stormwater drainage systems are often designed for peak flows associated with 10 to 50 year return periods.

## 3.6 Scaling of inter-amount times across scales: multifractal analysis

As explained in section 2, log-log plots of statistical moments versus sampling scale can be used to study scaling behaviour of time series. In the following, we plotted the moments $\langle X_\lambda^q \rangle$ of order $q$ of IATs as a function of mean inter-amount scale $\Delta q$ (proportional to the inverse of the scaling ratio $\lambda$), on a log-log scale, for moments of order 0.6 to 2.4. We applied the same procedure for flow time series over the same range of equivalent scales. Figure 11 shows examples of log-log plots for flow

volumes and IATs for McAlpine Creek (gauge 750). They show that log-linear fits are better for IATs than for flows, especially for higher order moments; minimum $R^2$ values, that are associated with fits for higher order moments, are 0.9972 and 0.9993 for flows and IAT respectively.

Plots in figure 11 show stronger departures from linearity in the log-log plots for flows than for IATs, especially for higher order moments. Figures 11c and 11d illustrate this for log-log curves of moment $q = 2.4$, where a scale break was detected at 22 hours for flows and subtle departures from linearity were found at 20.4 days for flows as well as 11.2 hours and 17.3 days for IATs. Similar analyses were conducted for all gauges, table 5 summarises minimum $R^2$ values for log-moments fits for flows and IATs. Log-moments for IATs show near perfect fits for all gauges, with minimum $R^2$ values between 0.995 and

1.000. Quality of log-moments is consistently lower for all basins; minimum $R^2$ values are between 0.990 and 0.997, lower quality fits generally occuring for smaller basins. Investigation of departures from linearity showed that for flows, most gauges exhibited a scale break between 8 and 20 days. Similar scale breaks, between time scales of 8 to 16 days, were found in scaling analyses of flow data by other authors based on flow data at daily resolution (Tessier et al., 1996; Labat et al., 2002; Sauquet et al., 2008). Labat et al. (2013) and Sauquet et al. (2008) found scale breaks in the range of 16 to 27 hours, for 30 minutes

respectively hourly resolution. We did not detect any strong departures from linearity in the IAT framework except for the 3 gauges where low flow regulation is applied (LSugarA, 507, LSugarP, 530, Stewart Creek, 750).

Using the empirical log-moments, we fitted the multifractal parameters $C_1$ and $\alpha$ for IATs and flow amounts. Table 5 summarises $C_1$ and $\alpha$ values for all basins, for flows and for IATs. Results show that $C_1$- values, characterising intermittency of the time series, are lower for IATs than for flows. This makes sense and can be explained by the adaptive sampling strategy

of IATs, especially the fact that low flows are sampled less often than in the classical fixed-time framework. Values of the multi-fractality index $\alpha$ are generally lower for IATs, with the exception of four basins. Two of these basins are characterised by low flow regulation, one basin has anomalous land-use distribution with a high concentration of imperviousness in the upper part of the basin. Time series of the 4[th] basin is short (8 years), which might influence outcomes of the scaling analysis. $C_1$ and $\alpha$ values for flows are in the range of values found by other authors. Figure 12 shows scatter plots of values for $C_1$ and $\alpha$

for flow and for IATs versus basin size and imperviousness. $C_1$- values are clearly negatively correlated with basin area. Rank correlations for IATs are -0.67 and -0.85 for flows. No significant correlation of $C_1$ with imperviousness was found, but the three basins with low flow control stand out with lower than average $C_1$ values. This shows up both in the IAT analyses and in the classical approach based on flows. The $\alpha$ values for IATs are positively correlated with area (0.6) and negatively with imperviousness (-0.56). No significant correlation with area nor imperviousness was detected. For IATs, negative correlation of

$\alpha$ with imperviousness comes from the fact that IATs in highly impervious basins are redistributed more evenly when moving from large to small scales (due to high imperviousness).

## 4 Summary and conclusions

In this study, we introduced an alternative approach for analysis of hydrological flow time series, using an adaptive sampling framework based on inter-amount times (IATs). The main difference between flow time series and time series for IATs is the rate at which low and high flows are sampled; the unit of analysis for inter-amount times is a fixed flow amount, instead of a fixed time window. Thus, in IAT analysis, sampling rate is adapted according to the local variability in flow time series, as opposed to time series sampling using fixed time steps. We aimed to investigate the effect of adaptive IAT sampling on flow statistics, especially on the tails of the statistical distributions associated with peak flow and low flow extremes. We analysed and compared statistical distributions of flows and IATs across a wide range of sampling scales to investigate sensitivity of statistical properties such as distribution quantiles, variance, scaling parameters and flashiness indicators to the sampling scale. We did this based on streamflow time series for 17 (semi)urbanised basins in North Carolina, US. The following conclusions were drawn from the analyses:

1. Adaptive sampling of flow time series based on inter-amounts leads to higher sampling frequency during high flow periods compared to conventional sampling based on fixed time windows. This results in a more balanced representation of low flow and peak flow values in the statistical distribution. While conventional sampling gives a lot of weight to low flows, as these are most ubiquitous in flow time series, IAT sampling gives relatively more weight to high flow periods, when given flow amounts are accumulated in shorter time. As a consequence, IAT sampling gives more information about the tail of the distribution associated with high flows, while conventional sampling gives relatively more information about low flow values.

2. Statistical analysis of IATs and flows at the 24 hour scale showed that coefficient of variation (CV) and skewness values were much higher for flows than for IATs, while medcouple values were lower for flows, indicating strong asymmetry of the flow distributions and low representation of high flow extremes in the statistical distribution. Larger basins were generally characterised by stronger smoothing of flows, resulting in higher mean flow, lower CVs and lower skewness of the histograms. Flow variability was clearly correlated with basin size, which is mainly a result of smoothing of low flows, in the left tail of the flow histogram. Larger imperviousness was associated with higher mean flows and significantly lower CV-values for IATs, which was mainly associated with stronger flow regulation by dams and detention ponds in urbanised basins

3. Comparison of coeffients of variation (CV) across the 17 basins showed that CV values of flows were negatively correlated with basin size, due to an increased smoothing effect for larger basin areas on flow variation. CV values of IAT distributions were not significantly correlated with basin size, which indicates that basin size has a stronger smoothing effect on low flow variability, strongly represented in conventional flow time series, than on peak flows which are more frequently represented in IAT time series. By contrast, CV values of IAT distributions were negatively correlated with imperviousness, while correlation between CV values for flows and imperviousness was not significant. Negative correlation between IAT CV values and imperviousness probably indicates stronger influence of flow regulation in more

urbanised basins resulting in more uniform runoff during rainy periods. IATs during these periods concentrate relatively more closely to the mean and show fewer extremes. This result is contrary to findings in many other studies, where urbanisation tends to be associated with higher peak flows. In the basins analysed in this study, flow regulating measures as indicated by a higher number of dams for detention basins and high drainage connectivity associated flow capacity limitations explains the strong influence of flow regulation associated with urbanisation.

4. Histograms of first-order differences showed negative skewness for IATs and positive skewness for flows, for most of the basins, indicating prevalence of slow flow recession compared to flow rise. The three basins with low flow regulation could be recognised by their relatively low medcouple values (<0.4) for IAT differences, showing that hydrographs tend towards being symmetrical in these basins. Significant correlations were found between skewness and medcouple of IAT differences and imperviousness (Spearman correlations 0.75 and -0.55), showing that urbanisation is associated with more regulated flows, thus relatively more symmetrical hydrographs with flow rise and recession at similar rates and lower frequencies of steep flow rise. Here, subbasin correlation appears to play a role: medcouple values were higher overall in the McAlpine subbasins than in Little Sugar Creek and Irwin subbasins. No significant correlations were found for differences in flows.

5. Quantile plots of flows and IATs plotted over a range of subdaily to seasonal scales showed the influence of the different sampling strategy for IATs compared to conventional flow sampling on median, 25-75, 10-90 and 1-99 percentile ranges of the distributions. The 25-75 and 10-90 percentile ranges for flows remained approximately constant, but the distance between 90 and 99 percentile values rapidly increased towards smaller scales. This reflects the highly skewed nature of flow distributions caused by oversampling of low flows compared to high flows; an effect that increased progressively towards smaller scales. By contrast, 10-90 and 1-99 percentile ranges for IATs increased more or less similarly with scale, for sampling scales ranging from 0.51 mm to approximately 10-16 mm, largely associated with intra-event flow variability. This indicates that the tails of IAT distributions are more or less equally sampled, at least up to the 1 and 99 percentiles.

6. Quantile plots for IATs showed different scaling at small scales (up to inter-amount scale 8-10mm) and large scales (roughly exceeding 20 mm inter-amounts), with a transition range in between. At smaller scales, IATs are mostly dominated by intra-event variability, while at large-scales IATs span multiple events. Flows sampled over fixed time intervals did not clearly exhibit this transition, probably because peak flow variability is being poorly sampled by fixed time window sampling. Because IATs adapt the sampling rate depending on the level of activity, they still capture a fair amount of peak flow statistics and intra-event properties, even at coarser scales.

7. Comparison of the tails of flows and IAT distributions showed that the distribution tail associated with low flows captures lower flow extremes in conventional sampling than in IAT sampling (0.02 mm/day compared to 0.1 mm/day). Conversely, IAT distributions capture more extreme peak flow values than conventional flow sampling, at the same sampling scale: the 99 percentiles for flows are associated with peak flows of 0.38 to 0.78 mm/h (sampling scales 12 hours to 4 days),

while 1 percentiles of IATs are associated with peak flows of about 20 mm/h (sampling scales 0.5 to 4 mm inter-amounts, associated with IATs of 12 hours to 4 days).

8. Analysis of CV-values of flow and IAT distribution across scale showed that at smaller scales, CV-values for flows increase more rapidly than for IATs, indicating that IAT variance remains more stable at smaller scales, while variance rapidly increases at small scales for flows. This is as a result of growing skewness of the statistical distribution of flows, caused by relative oversampling of low flows, or conversely, undersampling of high flows. This shows that for analysis of peak flows, IAT analysis offers advantages of the fixed-time sampling framework, as it samples peak flows more frequently and results in stabler variance across scales. For analysis of low flows, especially in basins characterised by strongly smoothed flow variability, IAT analysis offers little advantage and convential flow statistics are more suitable.

9. An IAT flashiness indicator was defined as the inter-amount scale at which 1% of flow accumulations occur in less than 15 minutes. Comparison between IAT-based flashiness and the commonly applied R-B flashiness index showed that indices were moderately correlated (Spearman rank correlation 0.55), yet there were some striking differences. R-B flashiness was shown to be strongly sensitive to low flow variability, while IAT flashiness was more sensitive to occurrence of peak values. Both flashiness indices showed strong correlation with basin area. R-B flashiness showed no clear relationship with imperviousness. IAT flashiness tends to decrease for a combination of higher imperviousness and larger basin size, basin size playing a stronger role than urbanisation. The effect of urbanisation on flow patterns for the basins in the study area is a mixture of faster runoff flows due to imperviousness and stronger flow regulation by dams and detention basins. This leads to a mixed effect of basin size, imperviousness and flow regulation on IAT flashiness and peak flows.

10. A minimum observable inter-amount scale was defined as the smallest scale at which flow variations can be studied given a fixed temporal observational resolution. At higher sampling scales, a growing percentage of flow accumulations occurs in less than the given observational resolution, 15 minutes in this study. This typically coincides with peak flows and implies that during peak events, the observational resolution is too low to measure flow variability. IAT analysis can thus be used to identify a critical resolution for flow observations, if a given peak flow accumulation is of interest. If correct observation of peak flows of a given magnitude is important, flow data need to be collected at a higher than 15 minute resolution during times of peak flows. This is typically the case of urban basins, where stormwater drainage systems are often designed for peak flows associated with 10 to 50 year return periods.

11. Multifractal analysis of IATs and flows was applied over a range of sub-daily to seasonal scales. Flows exhibited departures from multifractality for most basins, while IATs systematically scaled better than flows and showed departures from multifractality only for three basins subject to low flow regulation. This showed that IATs can help better predict peak flow characteristics at small unobservable scales based on coarse resolution data. Additionally, they provide new interesting alternatives for the stochastic modelling and downscaling of flow data.

This study showed that properties of statistical distributions of flow time series are very sensitive to the scale at which the statistics have been derived. This influences values of summary statistics that are used to characterise flow patterns of

hydrological basins, like peak flows at given recurrence intervals and flashiness indices. Adaptive sampling based on inter-amount times helped to achieve stabler variance across scales, yet the behaviour of other statistical properties such as skewness, medcouple is less clear. Further investigations are needed to interpret changes of statistics across scales. Future work will focus on multi-scale analysis, on how to compare results at different scales and what can be learnt from behaviour at different scales

about flow variability in hydrological basins in relation to basin characteristics.

Analyses in this study identified minimum observable scales below which flow variability cannot be captured at a given measurement resolution. The combination of being able to identify these minimum observable scales and to downscale flow data based on IATs is an interesting area for future investigation. Results showed that scaling parameters for IAT time series were more reliable than those based on fixed-time sampling because of smaller departures from linearity in log-log plots.

Future work will focus on possible ways to use IATs to downscale coarse resolution flow data with the help of multifractals and multiplicative random cascades, to see if this leads to more robust and reliable results than downscaling based on conventional flow time series.

Another aspect that remains to be investigated is how IATs computed on flow data compare to IATs computed on associated rainfall time series. Because flow is linked to rainfall, the comparison of the two could help better distinguish which aspects of

flow variability are due to rainfall and which relate to basin characteristics and stormwater management.

*Acknowledgements.* The authors would like to acknowledge USGS for making available the datasets of flow gauges in Charlotte. The first author would like to thank NWO Aspasia and Delft University of Technology for the grant that supported this research collaboration. The second author acknowledges the funding provided by the Swiss National Science Foundation, grant P300P2_158499 (project STORMS).

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

List of tables and figures

Table 1: Summary of results reported on the literature for (multi)fractal analysis of hydrological flows. MA: moments analysis, MFA: multifractal analysis, SA: Spectral analysis, TMA: Trace Moment analysis

Table 2: Summary of hydrological basins in the Charlotte area: basin area [km$^2$], imperviousness [%], average 24 h flow [m$^3$], average 24 h flow normalized by basin area [mm] and length of observation in years.

Table 3: Summary statistics of time series for flows and inter-amount times, at 24 hour sampling scale: coefficient of variation (CV), skewness (skew) and medcouple (mc).

Table 4: Summary statistics of time series for flows and inter-amount times, at 24 hour sampling scale: coefficient of variation (CV), skewness (skew) and medcouple (mc), for three sets of connected subbasins in the Charlotte catchments: Irwin, Little Sugar and McAlpine

Table 5: Minimum and maximum observable scales (in hours), flashiness index for 15 min observation time (in hours) and fitted multifractal parameters $\alpha$ and $C_1$ for inter-amount times respectively flows

Figure 1: Map with the location of the area, boundaries of hydrological basins and location of stream gauges used in the analysis (NB use figure1 from Joyce's paper: 1a + 1g)

Figure 2a, 2b: Example of times series for flow (2a) and for associated inter-amount times (2b) for the flow gauge at Taggart Creek, a 13.6 km2 basin in the Charlotte catchment.

Figure 3: Illustration of inter-amount data sampling for cumulative flow over a period of 7 days, for Taggart Creek. 3a Flow data series at original 15 min observational resolution; 3b Cumulative graph for flows and IATs at the same mean sampling resolution, illustrating how adaptive sampling based on IATs differs from classical fixed-time sampling)

Figure 4a, 4b: Histograms of flow time series (4a) and time series of inter-amount times (4b) for Taggart Creek and Little Sugar Creek at Archdale (LSugarA), for 24 hour scale.

Figure 5a-f: Scatter plots for mean normalised flows inter-amounts (a, b), coefficient of variation (c, d) and medcouple values (e, f) for flows and inter-amount times versus basin area and imperviousness degree. Grey triangle symbols represent inter-amount times, black circles represent flows.

Figure 6: histograms of first-order differences in inter-amount times and flows, at 24 hour sampling scale, for Irvins Creek and LSugarM Creek

Figure 7a-d: Scatter plots of skewness (a, b) and medcouple values (c, d) of histograms for differences in flows and inter-amount times, plotted versus basin size and imperviousness degree. Grey triangle symbols represent inter-amount times, black circles represent flows.

Figure 8a, 8b: Quantile plots of flows (a) and inter-amount times (b) for Taggart Creek for a range of scales, from 12 hours to 60 days. The bold black line denotes the mean values. The dotted black line shows median values. The central part of boxplots represents the 25-75 percentile range, upper and lower whiskers 10-90 percentile range, crosses the 1-99 percentile range.

Figure 9: Coefficients of variation for flows and inter-amount times scales across a range of sub-daily (3 to 12 hours) up to bi-monthly (60-68 days) scale, for Irvins Creek, LSugarM, Stewart Creek and McAlpine

Figure 10: Scatter plots of flashiness versus basin area and imperviousness, for all gauges

Figure 11: Example of log-log plots for flows and inter-amount times (11a, 11b), for Mc Alpine Creek, illustrating departures from linearity at high order moments. Curve for moment q=2.4 illustrating scale breaks for flows and inter-amount times (11c, 11d)

Figure 12: Multifractal parameters C1 and alpha for scaling analysis of flows and inter-amount times, as a function of drainage area and imperviousness degree

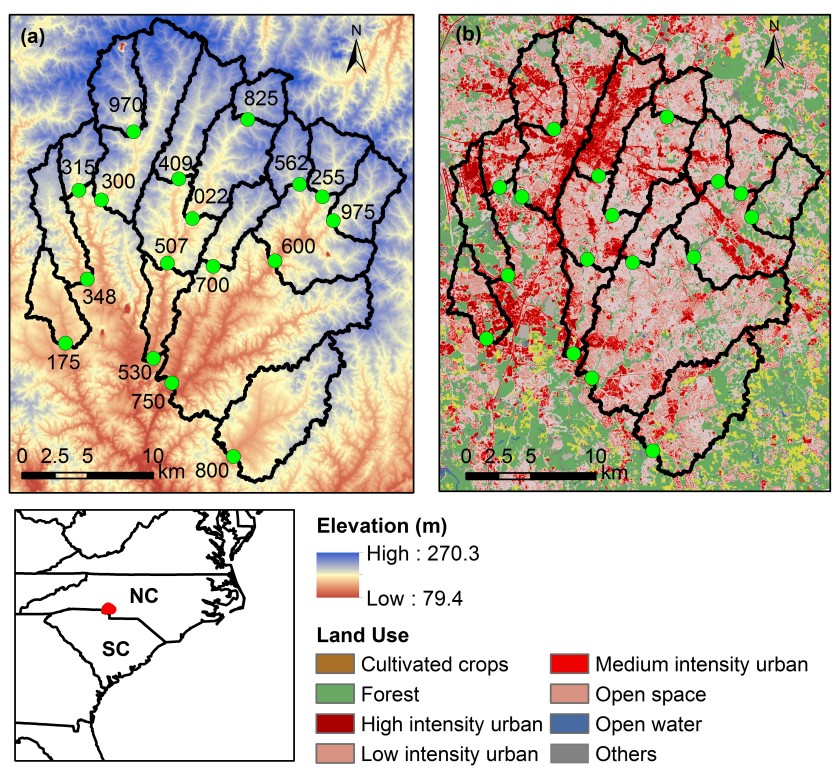

**Figure 1.** Map with the location of the area, boundaries of hydrological basins and location of stream gauges used in the analysis

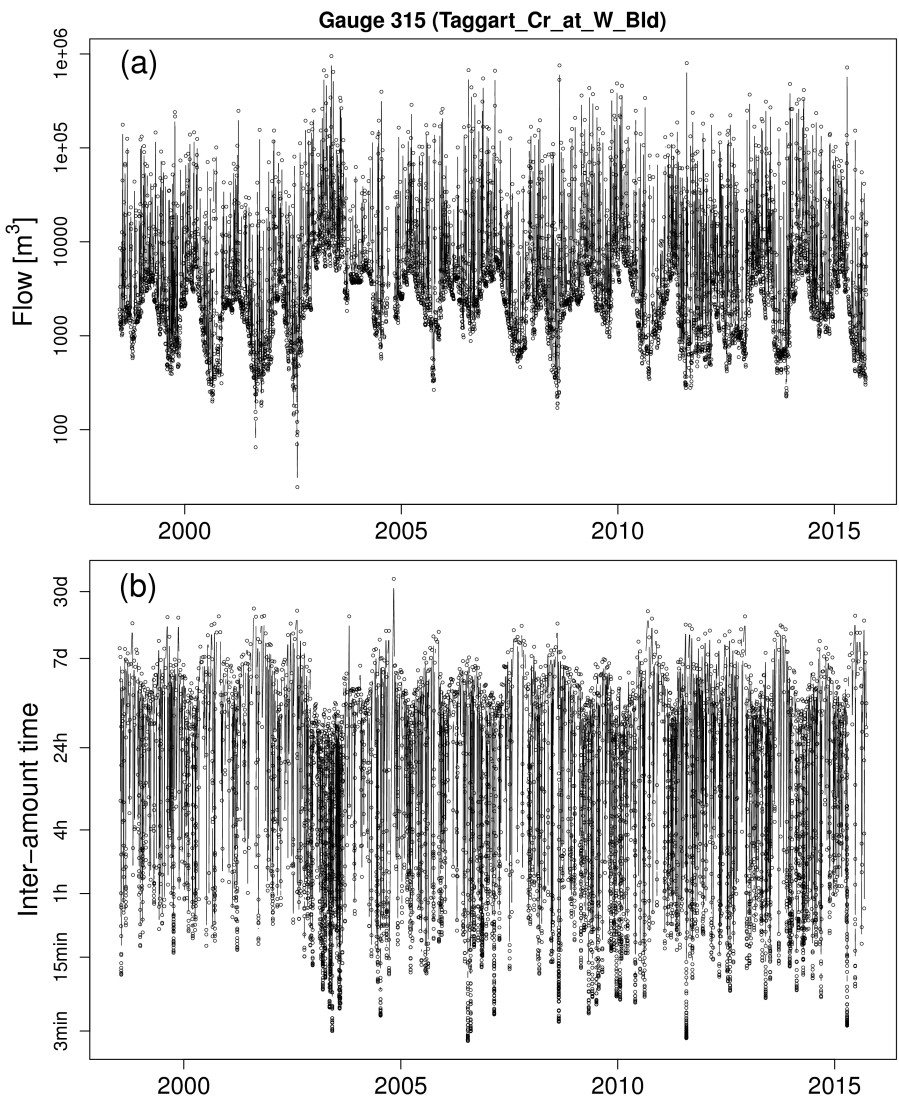

**Figure 2.** Example of times series for flow (a) and for associated inter-amount times (b) for the flow gauge at Taggart Creek, a 13.6 km$^2$ basin in the Charlotte catchment.

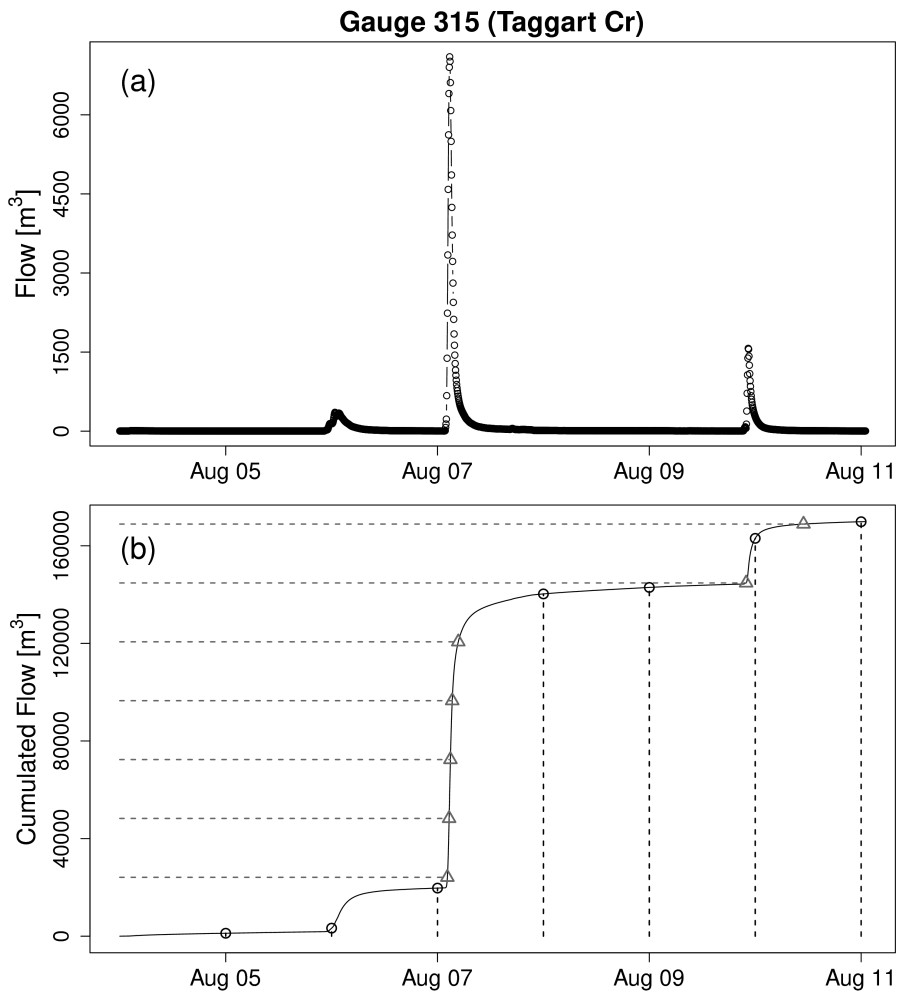

**Figure 3.** Illustration of inter-amount data sampling for cumulative flow over a period of 7 days, for Taggart Creek. 3a Flow data series at original 15 min observational resolution; 3b Cumulative graph for flows and IATs at the same mean sampling resolution, illustrating how adaptive sampling based on IATs differs from classical fixed-time sampling.

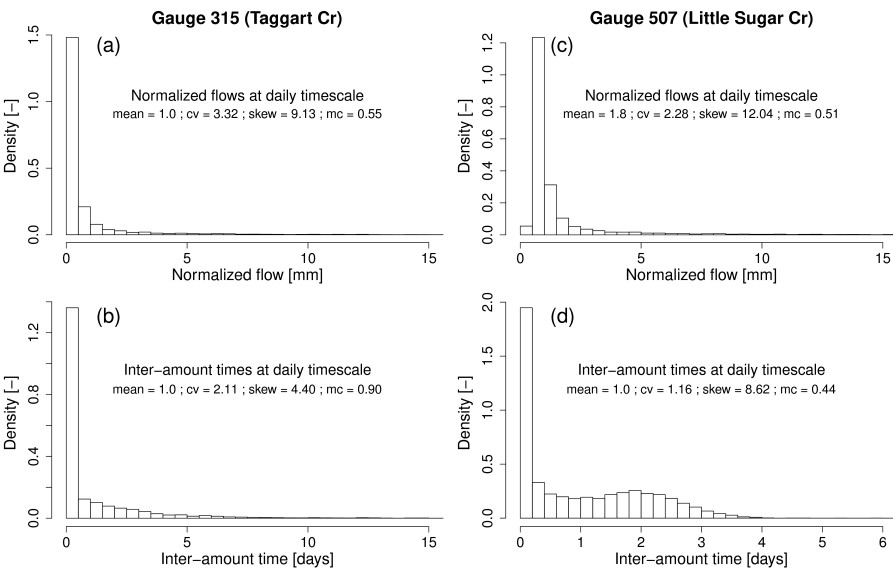

**Figure 4.** Histograms of flow time series (a) and time series of inter-amount times (b) for Taggart Creek and Little Sugar Creek at Archdale (LSugarA), for 24 hour scale.

**Table 1.** Summary of results reported on the literature for (multi)fractal analysis of hydrological flows. MA: moments analysis, MFA: multifractal analysis, SA: Spectral analysis, TMA: Trace Moment analysis

| Reference | Method | Sampling scale | Basins | Time series length | Scale break break | Value C1 | Value alpha |
|---|---|---|---|---|---|---|---|
| Tessier et al. (1996) | MFA | day | 30 basins in FR 40-200 km$^2$ | 11-30 yrs | 16 days 16 days | 1-16d: 0.2 +/- 0.1 30-4096d: 0.2 +/- 0.1 | 1-16d: 1.45 +/-0.25 30-4096d: 1.45 +/- 0.2 |
| Sauquet et al. (2008) | SA | Hour | 34 basins in FR | 16-37 yrs | 8.7h-7 d | - | - |
| Sauquet et al. (2008) | MA | Hour | 12.7-703 km$^2$ | 16-37 yrs | 10h-6.25d[*] | - | - |
| Sauquet et al. (2008) | SA | Day | Idem | Idem | 12 days | - | - |
| Pandey et al. (1998) | SA | Day | 19 basins USA 5 -1.8 10$^6$ km$^2$ | 9-73 yrs 9-73 yrs | 8 days 8 days | 1-8d: 0.2 +/-0.1 1-8d: 0.2 +/-0.1 | 1-8d: 1.65 +/0.12, 1-8d: 1.65 +/0.12 |
| Labat et al. (2013) | SA | 30 min | 3 basins in FR | - | 1 day | - | - |
| Labat et al. (2013) | TMA | 30 min | ca. 13 km$^2$ | - | 16 hours | 30min-16h: 0.22 >16h: 0.35 | 30min-16h: 1.18 >16h: 0.79 |

[*] only for higher order moments

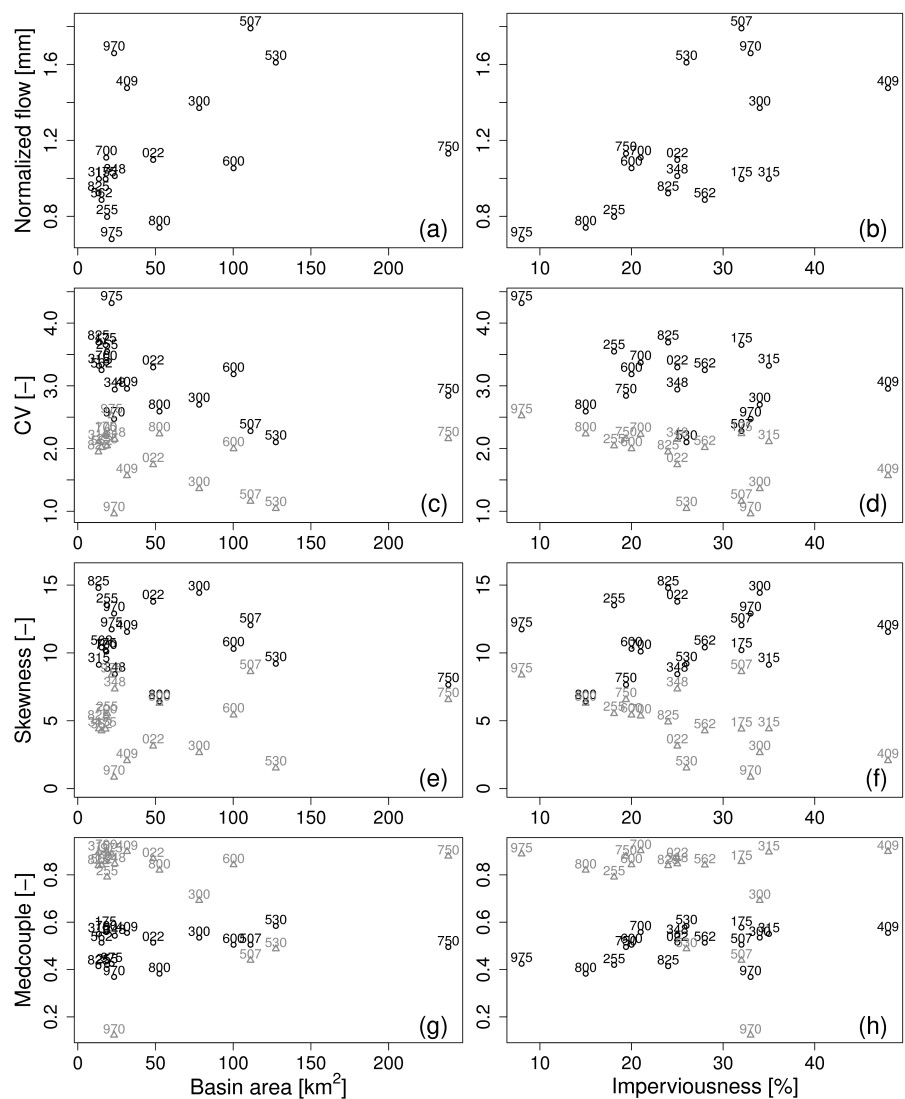

**Figure 5.** Scatter plots for mean normalised flows inter-amounts (a, b), coefficient of variation (c, d) and medcouple values (e, f) for flows and inter-amount times versus basin area and imperviousness degree. Grey triangle symbols represent inter-amount times, black circles represent flows.

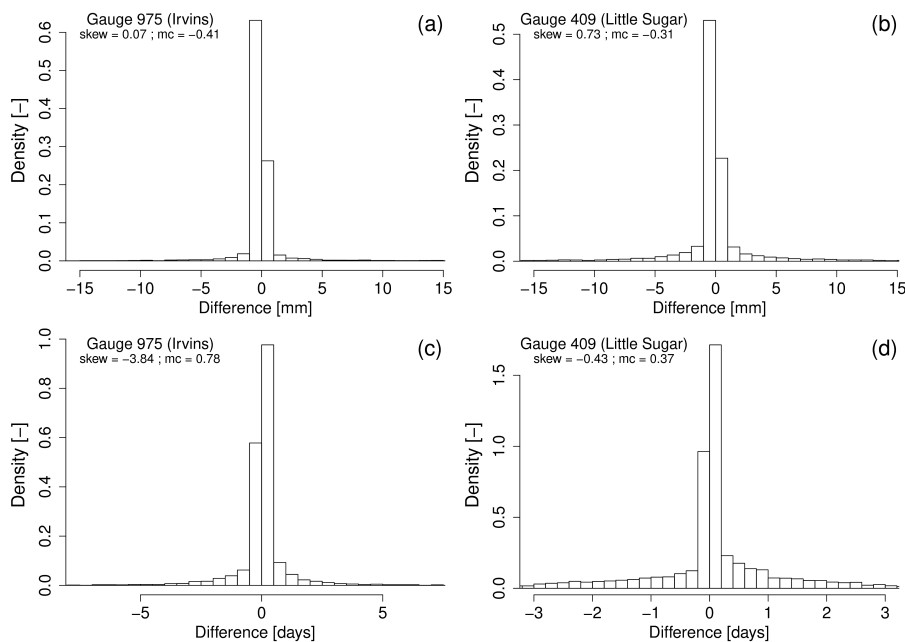

**Figure 6.** histograms of first-order differences in inter-amount times and flows, at 24 hour analysis sampling scale, for Irvins Creek and LSugarM Creek.

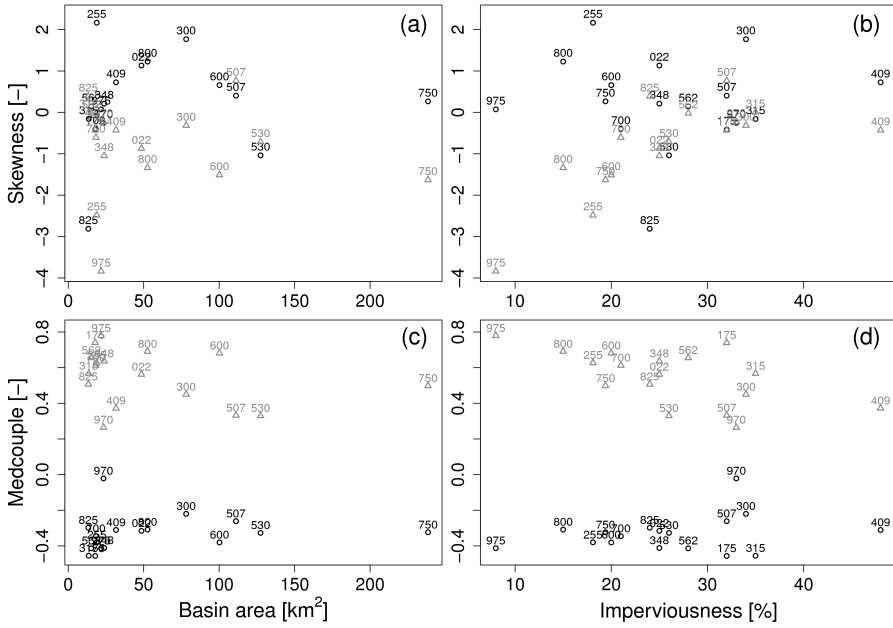

**Figure 7.** Scatter plots of skewness (a, b) and medcouple values (c, d) of histograms for differences in flows and inter-amount times, plotted versus basin size and imperviousness degree. Grey triangle symbols represent inter-amount times, black circles represent flows.

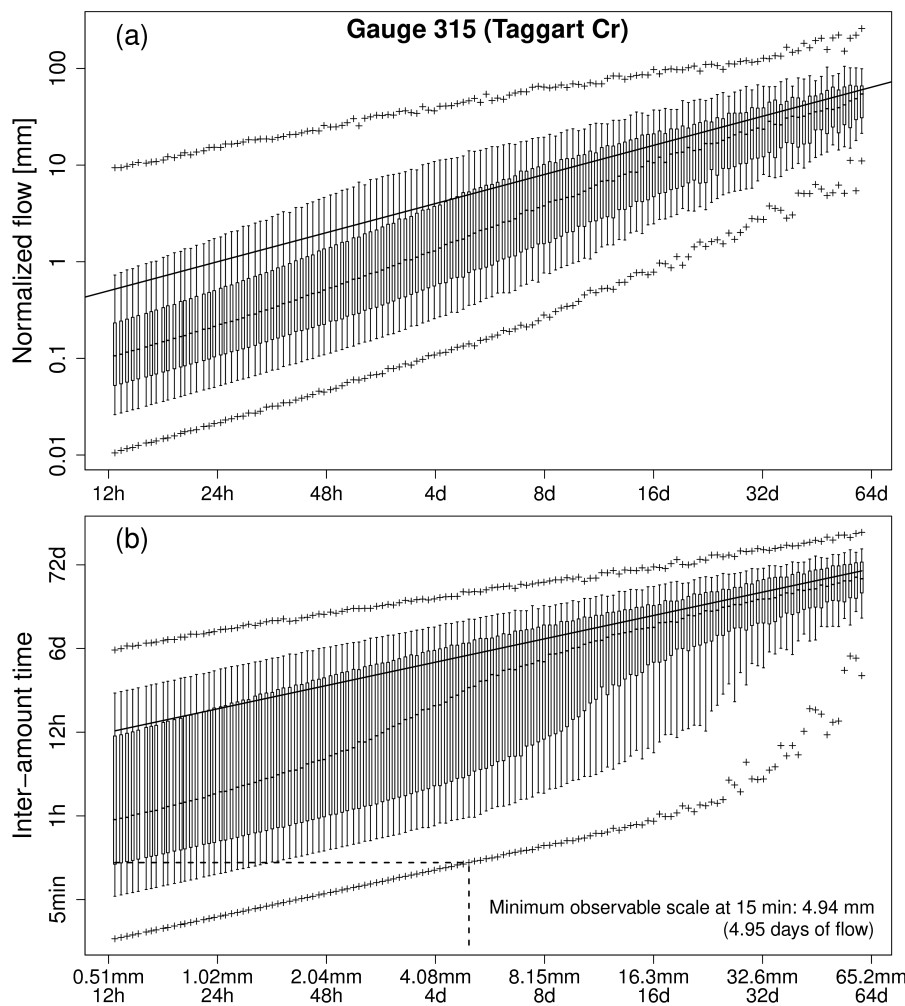

**Figure 8.** Quantile plots of flows (a) and inter-amount times (b) for Taggart Creek for a range of scales, from 12 hours to 60 days. The bold black line denotes the mean values. The dotted black line shows median values. The central part of boxplots represents the 25-75 percentile range, upper and lower whiskers 10-90 percentile range, crosses the 1-99 percentile range.

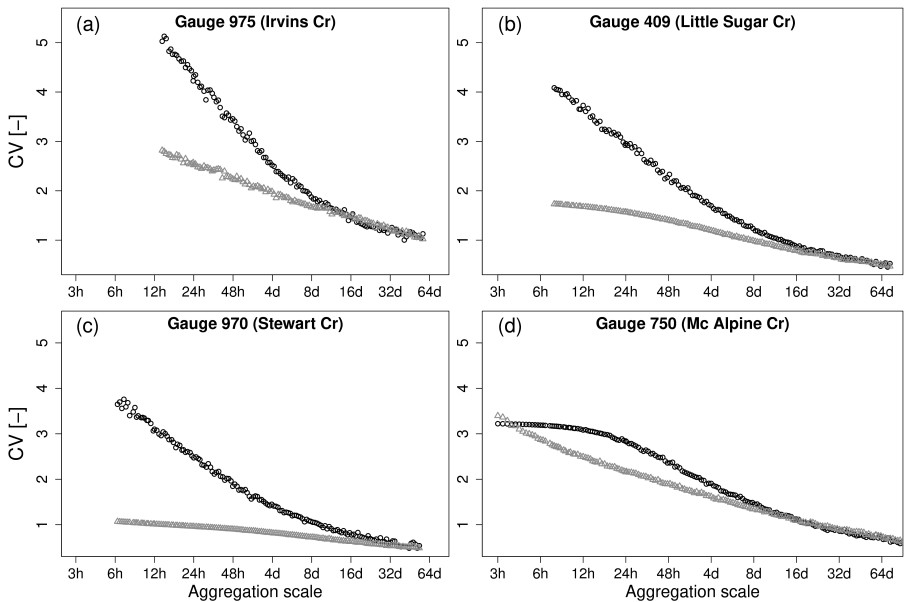

**Figure 9.** Coefficients of variation for flows and inter-amount times scales across a range of sub-daily (3 to 12 hours) up to bi-monthly (60-68 days) scale, for Irvins Creek, LSugarM, Stewart Creek and McAlpine. Grey triangle symbols represent inter-amount times, black circles represent flows.

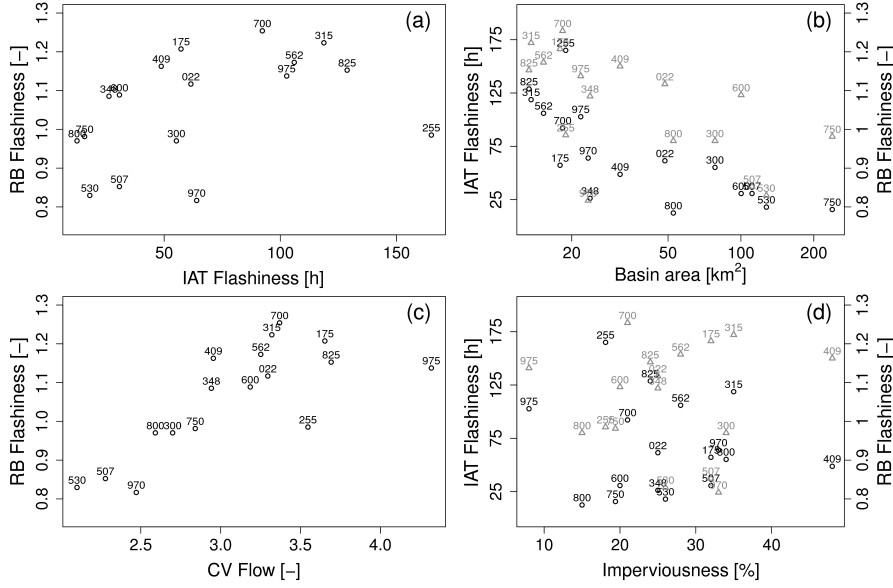

**Figure 10.** Scatter plots of flashiness versus basin area and imperviousness, for all gauges. Grey triangle symbols represent inter-amount times, black circles represent flows.

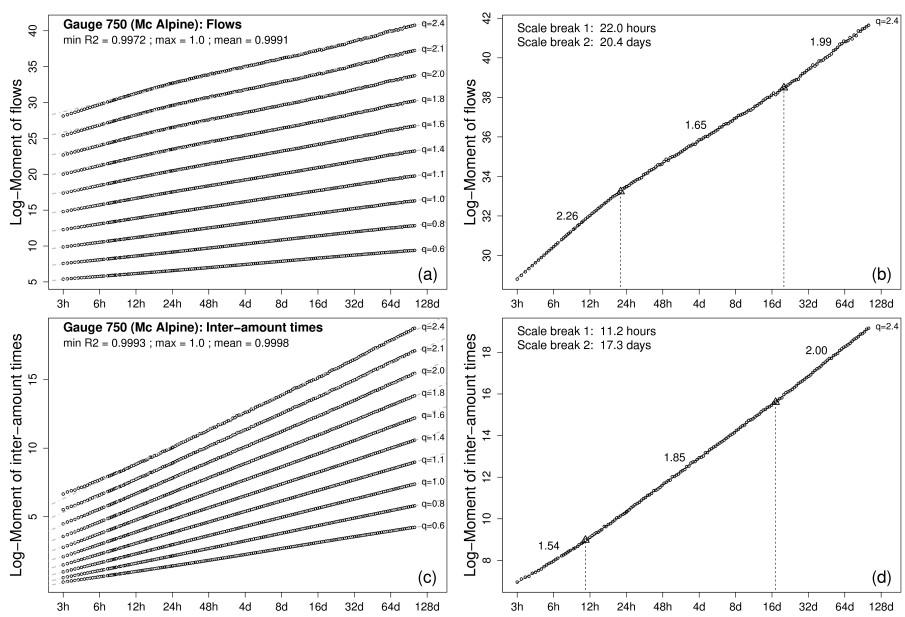

**Figure 11.** Example of log-log plots for flows and inter-amount times (a, b), for Mc Alpine Creek, illustrating departures from linearity at high order moments. Curve for moment q=2.4 illustrating scale breaks for flows and inter-amount times (c, d).

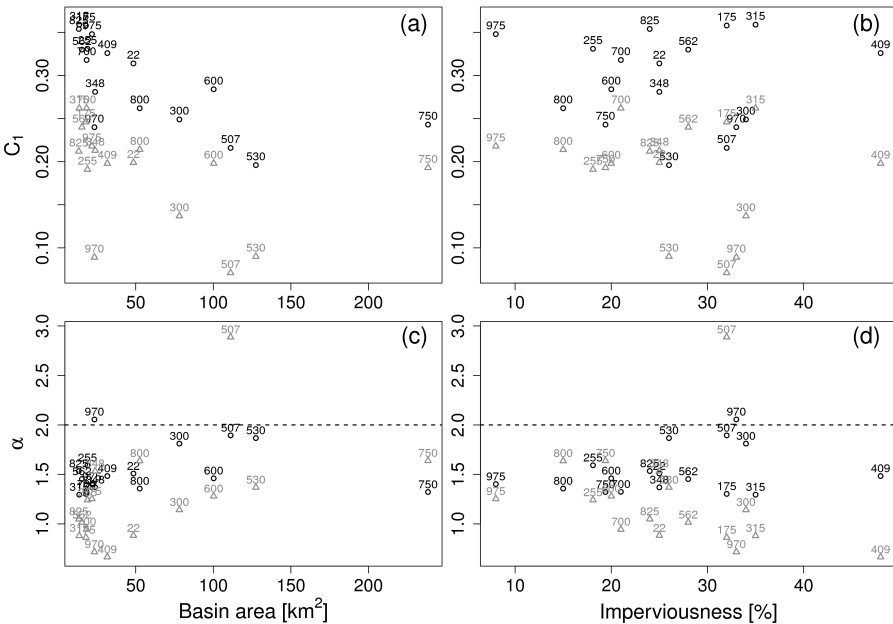

**Figure 12.** Multifractal parameters C1 and alpha for scaling analysis of flows and inter-amount times, as a function of drainage area and imperviousness degree. Grey triangle symbols represent inter-amount times, black circles represent flows.

**Table 2.** Summary of hydrological basins in the Charlotte area: basin area [km$^2$], imperviousness [%], average 24 h flow [m$^3$], average 24 h flow normalized by basin area [mm] and length of observation in years.

| ID | name | area | imperv | dams | mean flow | mean norm. flow | Nyears |
|-----|----------|-------|--------|------|-----------|-----------------|--------|
| 825 | UBriar | 13.3 | 24.0 | 22 | 12275 | 0.92 | 17.4 |
| 315 | Taggart | 13.6 | 35.0 | 3 | 13559 | 1.00 | 17.2 |
| 562 | Campbell | 15.3 | 28.0 | 48 | 13567 | 0.89 | 16.2 |
| 175 | Steele | 17.9 | 32.0 | 21 | 17838 | 1.00 | 17.4 |
| 700 | McMullen | 18.3 | 21.0 | 15 | 20348 | 1.11 | 29.0 |
| 255 | UMcAlpine | 18.9 | 18.1 | 100 | 15061 | 0.80 | 16.3 |
| 975 | Irvins | 21.8 | 8.0 | 62 | 14821 | 0.68 | 16.3 |
| 970 | Stewart | 23.4 | 33.0 | 55 | 38800 | 1.66 | 15.3 |
| 348 | Coffey | 23.8 | 25.0 | 72 | 24104 | 1.01 | 17.0 |
| 409 | LSugarM | 31.7 | 48.0 | 2 | 46775 | 1.48 | 21.0 |
| 022 | LBriar | 48.5 | 25.0 | 17 | 53246 | 1.10 | 19.8 |
| 800 | SixMile | 52.6 | 15.0 | -99 | 38914 | 0.74 | 8.0 |
| 300 | UIrwin | 78.1 | 34.0 | 39 | 107119 | 1.37 | 29.0 |
| 600 | MMcAlpine | 100.2 | 20.0 | 51 | 105640 | 1.05 | 29.0 |
| 507 | LSugarA | 111.1 | 32.0 | 24 | 199002 | 1.79 | 29.0 |
| 530 | LSugarP | 127.4 | 26.0 | -99 | 205202 | 1.61 | 18.3 |
| 750 | LMcAlpine | 238.4 | 19.4 | -99 | 269534 | 1.13 | 29.0 |

**Table 3.** Summary statistics of time series for flows and inter-amount times, at 24 hour sampling scale: coefficient of variation (CV), skewness (skew) and medcouple (mc).

| name | CV IAT | CV flow | skew IAT | skew flow | mc IAT | mc flow | skew dIAT | skew dflow | mc dIAT | mc dflow |
|------|--------|---------|----------|-----------|--------|---------|-----------|------------|---------|----------|
| UBriar | 1.95 | 3.69 | 4.91 | 14.79 | 0.84 | 0.41 | 0.39 | -2.81 | 0.51 | -0.30 |
| Taggart | 2.11 | 3.32 | 4.40 | 9.13 | 0.90 | 0.55 | 0.00 | -0.16 | 0.57 | -0.46 |
| Campbell | 2.02 | 3.25 | 4.26 | 10.40 | 0.84 | 0.51 | -0.02 | 0.15 | 0.66 | -0.41 |
| Steele | 2.24 | 3.65 | 4.39 | 10.21 | 0.86 | 0.58 | -0.43 | -0.41 | 0.74 | -0.46 |
| McMullen | 2.22 | 3.37 | 5.35 | 10.10 | 0.90 | 0.56 | -0.61 | -0.40 | 0.61 | -0.35 |
| UMcAlpine | 2.04 | 3.55 | 5.53 | 13.51 | 0.79 | 0.42 | -2.48 | 2.17 | 0.63 | -0.38 |
| Irvins | 2.52 | 4.32 | 8.37 | 11.74 | 0.89 | 0.42 | -3.84 | 0.07 | 0.78 | -0.41 |
| Stewart | 0.96 | 2.47 | 0.84 | 12.90 | 0.12 | 0.37 | -0.23 | -0.25 | 0.26 | -0.02 |
| Coffey | 2.15 | 2.94 | 7.34 | 8.44 | 0.85 | 0.54 | -1.05 | 0.21 | 0.64 | -0.41 |
| LSugarM | 1.57 | 2.95 | 2.06 | 11.55 | 0.90 | 0.55 | -0.43 | 0.73 | 0.37 | -0.31 |
| LBriar | 1.74 | 3.30 | 3.13 | 13.77 | 0.87 | 0.51 | -0.87 | 1.13 | 0.56 | -0.32 |
| SixMile | 2.23 | 2.59 | 6.29 | 6.42 | 0.82 | 0.38 | -1.34 | 1.23 | 0.69 | -0.31 |
| UIrwin | 1.36 | 2.70 | 2.65 | 14.43 | 0.69 | 0.53 | -0.32 | 1.77 | 0.45 | -0.22 |
| MMcAlpine | 2.00 | 3.19 | 5.42 | 10.30 | 0.84 | 0.50 | -1.51 | 0.66 | 0.68 | -0.38 |
| LSugarA | 1.16 | 2.28 | 8.62 | 12.04 | 0.44 | 0.51 | 0.77 | 0.40 | 0.33 | -0.26 |
| LSugarP | 1.04 | 2.10 | 1.52 | 9.20 | 0.49 | 0.58 | -0.71 | -1.04 | 0.33 | -0.33 |
| LMcAlpine | 2.16 | 2.84 | 6.56 | 7.65 | 0.88 | 0.50 | -1.63 | 0.27 | 0.50 | -0.32 |

**Table 4.** Summary statistics of time series for flows and inter-amount times, at 24 hour sampling scale: coefficient of variation (CV), skewness (skew) and medcouple (mc), for three sets of connected subbasins in the Charlotte catchments: Irwin, Little Sugar and McAlpine

| ID | name | CV IAT | CV flow | skew IAT | skew flow | mc IAT | mc flow | skew dIAT | skew dflow | mc dIAT | mc dflow |
|-----|------|--------|---------|----------|-----------|--------|---------|-----------|------------|---------|----------|
| 970 | Stewart | 0.96 | 2.47 | 0.84 | 12.90 | 0.12 | 0.37 | -0.23 | -0.25 | 0.26 | -0.02 |
| 300 | UIrwin | 1.36 | 2.70 | 2.65 | 14.43 | 0.69 | 0.53 | -0.32 | 1.77 | 0.45 | -0.22 |
| 825 | UBriar | 1.95 | 3.69 | 4.91 | 14.79 | 0.84 | 0.41 | 0.39 | -2.81 | 0.51 | -0.30 |
| 022 | LBriar | 1.74 | 3.30 | 3.13 | 13.77 | 0.87 | 0.51 | -0.87 | 1.13 | 0.56 | -0.32 |
| 409 | LSugarM | 1.57 | 2.95 | 2.06 | 11.55 | 0.90 | 0.55 | -0.43 | 0.73 | 0.37 | -0.31 |
| 507 | LSugarA | 1.16 | 2.28 | 8.62 | 12.04 | 0.44 | 0.51 | 0.77 | 0.40 | 0.33 | -0.26 |
| 530 | LSugarP | 1.04 | 2.10 | 1.52 | 9.20 | 0.49 | 0.58 | -0.71 | -1.04 | 0.33 | -0.33 |
| 562 | Campbell | 2.02 | 3.25 | 4.26 | 10.40 | 0.84 | 0.51 | -0.02 | 0.15 | 0.66 | -0.41 |
| 255 | UMcAlpine | 2.04 | 3.55 | 5.53 | 13.51 | 0.79 | 0.42 | -2.48 | 2.17 | 0.63 | -0.38 |
| 975 | Irvins | 2.52 | 4.32 | 8.37 | 11.74 | 0.89 | 0.42 | -3.84 | 0.07 | 0.78 | -0.41 |
| 600 | MMcAlpine | 2.00 | 3.19 | 5.42 | 10.30 | 0.84 | 0.50 | -1.51 | 0.66 | 0.68 | -0.38 |
| 750 | LMcAlpine | 2.16 | 2.84 | 6.56 | 7.65 | 0.88 | 0.50 | -1.63 | 0.27 | 0.50 | -0.32 |

**Table 5.** Minimum and maximum observable scales (in hours), flashiness index for 15 min observation time (in hours) and fitted multifractal parameters $\alpha$ and $C_1$ for inter-amount times respectively flows.

| ID | min scale | max scale | flash | RB | min R2 IAT | min R2 flow | alpha IAT | alpha flow | C1 IAT | C1 flow |
|---|---|---|---|---|---|---|---|---|---|---|
| UBriar | 13.75 | 1462 | 128.75 | 1.15 | 0.999 | 0.994 | 1.05 | 1.53 | 0.21 | 0.35 |
| Taggart | 12.50 | 1443 | 118.75 | 1.22 | 0.999 | 0.993 | 0.88 | 1.30 | 0.26 | 0.36 |
| Campbell | 9.25 | 1360 | 106.00 | 1.17 | 1.000 | 0.993 | 1.01 | 1.45 | 0.24 | 0.33 |
| Steele | 9.50 | 1457 | 57.25 | 1.21 | 1.000 | 0.991 | 0.86 | 1.30 | 0.25 | 0.36 |
| McMullen | 11.00 | 2420 | 92.25 | 1.25 | 0.999 | 0.992 | 0.94 | 1.32 | 0.26 | 0.32 |
| UMcAlpine | 10.00 | 1367 | 165.00 | 0.99 | 1.000 | 0.990 | 1.24 | 1.59 | 0.19 | 0.33 |
| Irvins | 13.75 | 1367 | 102.75 | 1.14 | 0.999 | 0.991 | 1.25 | 1.40 | 0.22 | 0.35 |
| Stewart | 6.25 | 1284 | 64.00 | 0.82 | 1.000 | 0.994 | 0.72 | 2.06 | 0.09 | 0.24 |
| Coffey | 4.75 | 1422 | 26.25 | 1.09 | 0.999 | 0.997 | 1.53 | 1.37 | 0.21 | 0.28 |
| LSugarM | 7.50 | 1752 | 48.75 | 1.16 | 1.000 | 0.996 | 0.66 | 1.48 | 0.20 | 0.33 |
| LBriar | 6.75 | 1658 | 61.50 | 1.12 | 1.000 | 0.996 | 0.88 | 1.51 | 0.20 | 0.31 |
| SixMile | 3.00 | 672 | 12.50 | 0.97 | 0.999 | 0.995 | 1.64 | 1.36 | 0.21 | 0.26 |
| UIrwin | 5.00 | 2420 | 55.25 | 0.97 | 1.000 | 0.995 | 1.14 | 1.81 | 0.14 | 0.25 |
| MMcAlpine | 5.50 | 2420 | 30.75 | 1.09 | 1.000 | 0.996 | 1.28 | 1.46 | 0.20 | 0.28 |
| LSugarA | 3.50 | 2420 | 30.75 | 0.85 | 0.995 | 0.996 | 2.89 | 1.89 | 0.07 | 0.22 |
| LSugarP | 2.75 | 1532 | 18.00 | 0.83 | 1.000 | 0.996 | 1.37 | 1.87 | 0.09 | 0.20 |
| LMcAlpine | 3.00 | 2420 | 15.75 | 0.98 | 0.999 | 0.997 | 1.64 | 1.32 | 0.19 | 0.24 |