# Peer review of "Statistical analysis of hydrological response in urbanising catchments based on adaptive sampling using inter-amount times"

_Hydrology and Earth System Sciences, 2016_

## Referee Comment (RC1) · Anonymous Referee #1 · 24 Oct 2016

Review of "Characterising hydrological response in urban watersheds based on interamount time distributions" by ten Veldhuis and Schleiss

The study propose a methodology for sampling flow data sets from water courses in urban watersheds based on samples of equal volume.

**Data**

The stream gauge data used in this study has a temporal resolution of maximum 15 minutes (p4I17) but all results reported are for longer aggregation periods: in Figure 8 minimum of 12 hours and in Figure 9 and 10 a minimum of 3 hours. With data available at 15 minutes resolution it should be possible to detect rapid changes in the flow and

the manuscript seriously lack comparisons at the sub-hourly scales. One exception is in Table 3 where the flashiness index is reported for 15 min observational resolution; but here it is unclear whether the 99%-tile of the flow measurements at 15 minutes resolution would give the same answer. Please investigate this and add the results to the discussion. As illustrated by Figure 3 the inter-amount time methodology result in much less data points than the original data set, but it is very unclear why it provides a better data basis for discussing the hydrological properties than the original 15 minute resolution data.

"Missing data were treated as zeros" (p4l23): how does this influence the results and the estimation error?

The catchments used in the study are to some degree sub catchments of each other. This could mean that some stream gauges are correlated (e.g. 507 and 530 (Figure 1)) but this is not discussed in the manuscript. This should be discussed in general and specifically in relation to the results where all catchments are discussed (e.g. Tables 2 and 3 and Figures 5, 7, 20 and 13).

Figure 1 is very hard to read and could benefit from being enlarged. Also, some will not know where in the world Charlotte, NC, is and it would be beneficial to add a panel of the North American East Coast with a marking of where the study area is.

Figure 2 is not providing any insight and should be removed.

**Inter-amount times**

The definition of inter-amounts (p4l24-p5l2) is brief and to the point. The section on normalization of inter-amounts (p5ll3-12) is also brief and the arguments for the methodology are good. The section on sample estimate (p5l13-p6l18) is somewhat harder to follow. The section thoroughly explain how to convert a time series of flow measurements to a series of inter-amount times and the possible error introduced by the approach but in the results section a measure of the error associated with the present 15 minute resolution data and the present catchments is not reported. This is really needed as it should be really limited how important this is at this fine temporal resolution of the flow data.

CV, skewness and medcouple are used to compare inter-amount time and flow distributions. In general the discussion of the results (p9I5-p11I9) are for daily values. This section could be much more interesting by adding results for higher resolution since the native data resolution is so much higher than the daily scale. At p10II28-29 a bimodal histogram for catchments with low flow regulation is discussed but not shown, please add these in a supplement. From p10I30 to the end of the section is repetition that could and should be left out of the manuscript.

The distribution of changes in inter-amount times is used to identify rapidly increasing and decreasing trends (p11II10-30). Figure 6 summarizes the results (again at daily scale) between flow based and inter-amount time based investigations but I cannot see how there can be both inter-amount times and flows in the figure. And it is not clear how to quantitatively get more knowledge from the inter-amount times since the qualitative conclusions will be the same between flows and inter-amount times (even though the skew will be in opposite directions). Please elaborate on this and correct the figure.

Inter-amount times are further compared to flows in Figure 8. Figure 8 is a really good example of all the problems you get from having box-plots on a log scale. For both values span several orders of magnitude and is vastly skewed (as indicated by the large difference between the mean and the median). The associated discussion (p11l31-13l15) is very hard to follow and whether a given percentile is following a power law (p12ll10-12) or not is effectively impossible to see from the figure. I would suggest a form of normalization of the results to avoid the logarithmic axes in Figure 8 and make the scaling discussion much more accessible.

Flashiness

From the very first sentences of the abstract (p1II1-5) flashiness is highlighted as a key parameter where inter-amount time distributions can really make a difference. In the introduction it is concluded from literature that it is difficult to predict the flashiness of urban watersheds, but no methodologies or results are directly presented. Please add a more thorough introduction on how flashiness is normally calculated from traditional flow data.

The flashiness indicator formulated in the study (p6ll23-24) is very briefly described and a discussion of why this choice was made and why this is a good indicator for flashiness is completely lacking. Please add these.

The results and discussion for flashiness and minimum observable scale (p13l32 + p14) is not easy to follow. In the first section (p13l33-p14l10) It is discussed that very high peak flows cannot be measured correctly every time with 15 minutes resolution data; but is this interesting at all and is it important to know flow variation at this high resolution? Please add a discussion of this. In the next section ('Table 2' should really be 'Table 3') a clear correlation between minimum observable scale and flashiness is reported (p14ll18-20) as well as a clear correlation between flashiness and basin area (p14ll20-22 + Figure 10); these make perfect sense, but would they be different if flow-based flashiness indicators had been used? Please add a comparison to other flashiness indicators. In the very end of the section (p14ll30-34) a discussion of results not shown is given indicating that the manuscript would benefit from addition of a supplement containing results from all catchments and also the further analysis that has apparently been carried out.

**Multifractal analysis**

The whole section on scaling (p15ll1-29) could really be shortened to one sentence simply stating that scaling is great for both flow and inter amounts accompanied by the left side of Figure 11 unless you can show that there is a statistical significant better fit of one of them. Also the identified departures from linearity (p15ll6-88 and the right

side of Figure 11) should be statistically significant to be relevant for discussion. Please provide relevant statistics to support the conclusions drawn or shorten the section.

**Conclusions**

In the conclusions it is stated that: "Flows sampled over fixed time intervals did not clearly exhibit this transition. This is result of peak flow variability being poorly sampled by fixed time window sampling." (p16ll28-29) but until you add results where you utilize the 15 minutes resolution this cannot be concluded.

Another sentence: "Based on inter-amount times distribution we were able to define a flashiness indicator that incorporates both the rising and falling components of the hydrological response" (p17ll28-29) seems to be unsupported as the flashiness indicator, as I have understood it, really only tell how many hours of mean flow one can expect as peak flow within a given much shorter time frame (e.g. an indicator of 100 hours for 15 minutes inter-amount times mean that the 99%-tile peak flow is 400 times the mean flow) and how the rise and fall of the peaks are incorporated is not clear.

It is also concluded from the multifractal analysis that: "This showed that inter-amount times can help better predict peak flow characteristics at small unobservable scales based on coarse resolution data. Additionally, they provide new interesting alternatives for the stochastic modelling and downscaling of flow data." (p17II18-20) and "Scaling analysis showed that inter-amount times provide a promising way to better predict peak flow characteristics at small unobservable scales from coarse resolution data" (p17II31-32) but this was not discussed at all before in the manuscript and if it is true you should really add results to support this.

**Figures**

In general the figures need some work before publication. ć The fonts used are generally very small (e.g. the legend for Figure 1 which is unreadable when printed). ć The use of sub-figure numbering is inconsistent between text and figures (e.g. Figure

4c and d are not mentioned in the caption and for Figures 6, 8 and 9 the sub-plot labels are missing). ć In Figure 9 there is no marking of which color corresponds to which data set. ć Inconsistent use of 'IATs' (Figure 11) and 'inter-amount time' (Figure 8) as well as 'Flow' (Figure 2) and 'Amounts' (Figure 11) and 'medcouple' (Figure 7) and 'MC' (Figure 5). ć What are the units of the x-axes of Figure 6? ć Also put the unit directly on the x-axes of Figure 4 and not only in the text of the figure. ć For Figure 8 the x-axis seem confusing. For the inter-amount times the volumes are based on time and should be reported something like "0.51 mm (12h)" but for the flow plot the axis should only be time. Similarly for Figure 11 where the x-axes for flow and inter-amount times should be different.

**Concluding remark**

Indeed, this approach is very interesting as it generate data sets with higher sampling frequency when high flow occur and lower sampling frequency for low flow periods. This is nicely pointed out by the authors. However, from the reported results I am not convinced that the methodology adds so much to the field. It is also unclear how exactly the authors see that this knowledge can be utilized in future research as 1) very high resolution flow data is used in this study and 2) it is unclear how the results can be used to better describe watersheds with much coarser data available as it is already pointed out that considerable uncertainty is associated with estimation of the peak flow from 15 minutes resolution data; how will than then look if only daily data is available?. Also downscaling of coarse flow data is mentioned, but for the same reasons as just mentioned it is very unpredictable how this will work.

---

## Referee Comment (RC2) · Anonymous Referee #2 · 29 Nov 2016

This paper applies an existing framework (Schleiss and Smith, 2016) for analysing inter-amount-times (IAT) to time series of daily streamflow from 17 urbanised watershed monitored by the USGS. The stated aim of the paper is to explore how IAT can be used to characterize the hydrological response and compare results to analysis based on traditional analysis techniques.

In general I found the manuscript difficult to read for a number of reasons ć The introduction is very unstructured. It starts by discussing effects of urbanisation, then gives a general A-Z of general challenges in applied hydrology, incl. flood frequency analysis, flow duration curves, multivariate statistics, unit hydrographs, baseflow separation, before ending-up with a very detailed summary of scaling issues with a high

level of assumed knowledge on behalf of the reader. I suggest a much more focussed introduction is necessary to better justify the scientific gaps being addressed by the study.

ć The aim is presented very loosely as 'to explore how inter-amounts can be used to characterise hydrologic response for a range of (semi)urban watersheds in North Carolina, US.' I think it would be more convincing if the aim could be linked more strongly to a distinct problem/gap when using existing methods for analysing the hydrological response from urban watersheds. What is the actual scientific and/or applied problem being addressed here?

ć I was struggling with some of the technical description in Section 2. This is partly down to a number of key places where the notation is hard to follow (see more detailed comments below), but also because I could not follow how this method was going to add new insight that was not available from a direct analysis of runoff time series. Again, I think a more focussed and readable introduction might have been helpful here.

åĂć The results discussion (Section 3) is hard to follow. Maybe consider introducing more subsection and better explain how the different analysis comes together to answer the scientific aim, rather than at present where I get the impression it is a series of independent and individual analysis undertaken because it is possible.

àĂć The main conclusion seems to be that flashiness is related to watershed area, but that no link to urbanisation could be identified. While this is, of course, not the fault of the researchers it does perhaps suggest that the title of the paper (urban watersheds) is not really appropriate as no new insight into the effects of urbanisation seems to have been discovered.

ć I don't think the strong conclusion on page 17, line 21-22 is justified as it is not clear what characteristics is being referred to that could not have been established using conventional time series?

Detailed comments I think HESS uses British rather than US spelling, so catchment rather than watershed

Page 2, line 25: Define 'scale-break'

Page 3, line 5: Not sure what 'moments q[0.1-4]' signifies?

Eq. (4): this equation sums over qi, but in the lines above the sample flow time series is defined in terms of r (line 14). Should it be 'r' in Eq(4) or else 'qi' need to be defined somewhere

Eq (5): I don't understand the notation used in this equation. What does 'Qt,nq-nq' mean?

Eq (6): I don't know how to link this equation to Eq. (5) - I think some more explanation is required here.

Eq(7): 'T' is not defined anywhere?

Page 6, lines 23-24: Given that the flashiness is one of the main conclusions of the study, I think a more comprehensive description of the concept is required, for example include a conceptual figure.

Page 7, line 16: What is 'Scott's rule'?

Page 19=0, line 14: what does 'cq' refers to?

**C3**

---

## Author Comment (AC1) · 6 Dec 2016

Review of "Characterising hydrological response in urban watersheds based on interamount time distributions" by ten Veldhuis and Schleiss Referee #1 RC: referee comment; Reply: authors' reply.

Reply (general): Some of the review comments made us realise that the motivation for exploring inter-amount times (IAT) statistics for flow data needs better explanation. In order to make the context and motivation for IAT analysis more clear, we will restructure and focus the introduction section (see also reply to Reviewer#2) and rephrase the objective of our study.

Firstly, IAT analysis is not meant to replace conventional analyses based on flow data, rather to complement it. Secondly, the complementary value of IAT analysis lies in its different sampling strategy: sampling frequency for computation of IATs is adapted to the amount of flow that passes the sampling point. That is, more samples are taking during periods of high flow and fewer samples are taken during low flow conditions. As a result, statistical distributions give more weight to high flow data based in fixed to low flow periods) compared to statistics for conventional flow data based in fixed temporal sampling windows. Previous studies of IAT analysis for rainfall data have shown to provide additional insights to conventional rainfall analysis. The aim of our study was to explore what additional information can be derived from IAT sampling. Sentences have been added to more explicitly explain these aspects, see Replies to comments below.

RC: The study propose a methodology for sampling flow data sets from water courses in urban watersheds based on samples of equal volume.

RC: Data The stream gauge data used in this study has a temporal resolution of maximum 15 minutes (p4l17) but all results reported are for longer aggregation periods: in Figure 8 minimum of 12 hours and in Figure 9 and 10 a minimum of 3 hours. With data available at 15 minutes resolution it should be possible to detect rapid changes in the flow and the manuscript seriously lack comparisons at the sub-hourly scales.

Reply: This is a misunderstanding that we will address in the revised manuscript. The reason why longer aggregation periods were used is that sub-hourly scales cannot be analysed (directly) in the IAT framework. We refer to the definition of minimum observable scale in equation 9: to limit estimation errors of inter-amount times, we allowed an average interpolation error smaller than 50%. Only inter-amount results associated with smaller average interpolation errors were included in the analysis. As can be seen in table 3, minimum observable scales vary between 2.75 and 13.75 hours as a result of this requirement. One key aspect of IAT analysis is that sampling frequency is not fixed, but varies depending on flow variability. Because of this, the time between two

HESSD
consecutive samples can be very short, even at large scales. What we call the "scale" of analysis is nothing else than the average time between two consecutive samples. But it should be clear that during periods of high flows, much shorter IATs than 24 hours (e.g., 15 minutes or less) are observed. So one of the interesting aspects of IATs is precisely that it helps identify situations in which rapid changes in flow cannot be detected as a result of a too low observational resolution. We will add the following sentences to explain this more explicitly: Introduction (p3, l24): While conventional sampling of flow time series is based on fixed time windows, inter-amount times are based on an adaptive sampling strategy: time series are sampled at higher frequency during periods of high flow and gives relatively little weight to periods of low flow. Inter-amount times sampling results in statistical distributions with different properties compared to conventional flow sampling and can provide different insights that complement traditional analysis Introduction (p3, I34): We find that (...) in intervals, as sampling is based on accumulated flows, independent of the original observation interval. The minimum resolution at which inter-amount times can be reliably sampled depends on the magnitude of flow peaks in the relation to the observation time interval, as explained in section 2.3.

RC: One exception is in Table 3 where the flashiness index is reported for 15 min observational resolution; but here it is unclear whether the 99%-tile of the flow measurements at 15 minutes resolution would give the same answer. Please investigate this and add the results to the discussion.

Reply: The flashiness indicator that we derived in our IAT analysis defines the scale at which 1% of flow accumulations occurs in less than 15 minutes. At higher resolutions, a growing percentage of flow accumulations occurs in less than 15 minutes, hence cannot be detected at the given observational resolution of 15 minutes. Quantiles of conventional flow time series do not provide this information. The 99%-tile gives a flow value that was exceeded 1% of the time, thus tells something about occurrence of peak flows at the sampled resolution (or at the observed resolution, for the 15 minutes time-

**HESSD**
scale). By nature of the sampling strategy this does not tell us to what extent observed values are able to capture flow variability in between sampling times.

RC: As illustrated by Figure 3 the inter-amount time methodology result in much less data points than the original data set, but it is very unclear why it provides a better data basis for discussing the hydrological properties than the original 15 minute resolution data.

Reply: This is a misunderstanding that we will clarify better in the figure caption: 3a shows original observed time series, 3b compares flow data and inter-amount data at a same resolution to illustrate how sampling strategies differ.

RC: "Missing data were treated as zeros" (p4l23): how does this influence the results and the estimation error?

Reply: As reported, the percentage of missing data was smaller than 5% for all gauges. Their effect on IATs is difficult to predict as this depends on the pattern of missing values and whether or not they occur during a period of low or peak flow. Sensitivity studies by Schleiss and Smith (2016) have shown that the general effect of replacing missing values by zeros is that a few sample IATs will be overestimated. This mostly affects the right tail of the distribution and tends to have limited impact on peak flow characteristics. We feel this is not a major issue here because (a) there are relatively few missing values and (b) by treating them as zeros, we always assume the worst case scenario.

RC: The catchments used in the study are to some degree sub catchments of each other. This could mean that some stream gauges are correlated (e.g. 507 and 530 (Figure 1)) but this is not discussed in the manuscript. This should be discussed in general and specifically in relation to the results where all catchments are discussed (e.g. Tables 2 and 3 and Figures 5, 7, 20 and 13).

Reply: The reviewer is right in that correlation between basin response may play a role,
through connections in the basin network and in the case of large storm events affecting multiple basins in the region. Previous studies have also shown that these relations are vastly complex, especially in urbanised basins. Spatial distribution of rainfall, flowpath network structure and regulation by stormwater management infrastructure all play a role. The interest of this analysis is in identifying and characterising differences and similarities in response between basins, which could help to gain better insight into where to look for possible hydrologic/hydraulic explanations, basin-to-basin response correlation being only one of the many possible factors.

RC: Figure 1 is very hard to read and could benefit from being enlarged. Also, some will not know where in the world Charlotte, NC, is and it would be beneficial to add a panel of the North American East Coast with a marking of where the study area is.

Reply: thanks for this comment, we will make sure to improve the figure.

RC: Figure 2 is not providing any insight and should be removed.

Reply: Figure 2 visualises times series of flows and associated inter-amount times and shows how the inter-amount times have lower variability and fewer outliers as a result of the different sampling strategies. These aspects are discussed later in relation to the respective statistical distributions. Since many readers will never have seen IAT time series we would prefer to keep this figure.

RC: Inter-amount times The definition of inter-amounts (p4l24-p5l2) is brief and to the point. The section on normalization of inter-amounts (p5ll3-12) is also brief and the arguments for the methodology are good. The section on sample estimate (p5l13-p6l18) is somewhat harder to follow. The section thoroughly explain how to convert a time series of flow measurements to a series of inter-amount times and the possible error introduced by the approach but in the results section a measure of the error associated with the present 15 minute resolution data and the present catchments is not reported. This is really needed as it should be really limited how important this is at this fine temporal resolution of the flow data.

HESSD
Reply: The error in flow measurement associated with temporal observational resolution, in relation to the real flow variability, is not known for these datasets. For this, a set of observations at higher temporal resolution would be required, for comparison. As reported in the paper, flow data are derived from water depth measurements using flow-rating curves that regularly recalibrated by USGS. Documentation reporting quantitative error estimates is not available to the authors.

RC: CV, skewness and medcouple are used to compare inter-amount time and flow distributions. In general the discussion of the results (p9I5-p11I9) are for daily values. This section could be much more interesting by adding results for higher resolution since the native data resolution is so much higher than the daily scale.

Reply: see previous comment, the resolution of IAT analysis is limited by our definition of acceptable error in IAT estimation. We agree that IAT analysis at higher resolution would be very interesting, but this would require much higher observational resolution during peak flow periods. Conversely, our IAT analysis shows that conventional analysis of flow data at 15 minute resolution during periods of peak flows is flawed, since variability in flow extremes cannot be captured at this resolution.

RC: At p10ll28-29 a bi-modal histogram for catchments with low flow regulation is discussed but not shown, please add these in a supplement.

Reply: An example of a bi-modal histogram is shown for basin 409, in figure 4d. We will make sure to add histograms for the other 2 basins (507, 970) as a supplement

RC: From p10l30 to the end of the section is repetition that could and should be left out of the manuscript.

Reply: this is a brief summary of the section and is meant to help the reader recap the main messages of this section. Given the complexity of the analysis and the fact that few readers will be familiar with the notion of inter-amount times, we would prefer to keep this paragraph.
RC: The distribution of changes in inter-amount times is used to identify rapidly increasing and decreasing trends (p11ll10-30). Figure 6 summarizes the results (again at daily scale) between flow based and inter-amount time based investigations but I cannot see how there can be both inter-amount times and flows in the figure. And it is not clear how to quantitatively get more knowledge from the inter-amount times since the qualitative conclusions will be the same between flows and inter-amount times (even though the skew will be in opposite directions). Please elaborate on this and correct the figure.

Reply: The reviewer is right, histograms for flows were missing. We apologise for this mistake and will make sure it is corrected The reviewer asks what more knowledge we gain from differences in inter-amount times compared to flows. We will look into this during the revisions and provide more quantitative conclusions or if indeed there is insufficient new quantitative insight, we will remove it.

RC: Inter-amount times are further compared to flows in Figure 8. Figure 8 is a really good example of all the problems you get from having box-plots on a log scale. For both values span several orders of magnitude and is vastly skewed (as indicated by the large difference between the mean and the median). The associated discussion (p11I31-13I15) is very hard to follow and whether a given percentile is following a power law (p12II10-12) or not is effectively impossible to see from the figure. I would suggest a form of normalization of the results to avoid the logarithmic axes in Figure 8 and make the scaling discussion much more accessible.

Reply: The reviewer is correct in that the values span 4 orders of magnitude in scale, hence our choice for visualising the Q-Q plots on a log-scale. Normalisation as suggested by the reviewer has been applied, the current figures are based on normalised values. The distributions are indeed vastly skewed; this is a property of the data and is not associated with visualisation on the log-scale. We will add text to the discussion of the figure, explaining that straight lines on the log plots indicate power-law scaling.

RC: Flashiness From the very first sentences of the abstract (p1ll1-5) flashiness is

**HESSD**
highlighted as a key parameter where inter-amount time distributions can really make a difference. In the introduction it is concluded from literature that it is difficult to predict the flashiness of urban watersheds, but no methodologies or results are directly presented. Please add a more thorough introduction on how flashiness is normally calculated from traditional flow data.

Reply: the reviewer has a valid point, we will add a brief literature review on flashiness indicators to the introduction. The most commonly used is the R-B flashiness index, introduced by Baker (2004). We will compute this index for our datasets and compare results with those for the flashiness indicator derived from IAT analysis. An important difference is that the R-B Index essentially measures variance, while the IAT flashiness focuses on high flow accumulations compared to the mean.

RC: The flashiness indicator formulated in the study (p6ll23-24) is very briefly described and a discussion of why this choice was made and why this is a good indicator for flashiness is completely lacking. Please add these.

Reply: the reviewer is right in that an explicit definition of the flashiness indicator is missing. We will add a clear definition and propose to the replace the existing text (p6, I 26-27) by: "In this work, we defined a flashiness indicator based on Inter-amount times, as follows: Def: inter-amount scale at which 1%-tile Inter-amount time falls below the observational resolution of 15 minutes."

RC: The results and discussion for flashiness and minimum observable scale (p13l32 +p14) is not easy to follow. In the first section (p13l33-p14l10) It is discussed that very high peak flows cannot be measured correctly every time with 15 minutes resolution data; but is this interesting at all and is it important to know flow variation at this high resolution? Please add a discussion of this.

Reply: We will rephrase the text to make this point more clear. Essentially, IAT analysis shows at which point flow accumulation within the 15 min observational time window is too high to be properly measured. This is relevant, because it shows that high flow
values occur that are not observed at the given measurement resolution. These values could include critical peak values, for instance for flood generation.

RC: In the next section ('Table 2' should really be 'Table 3') a clear correlation between minimum observable scale and flashiness is reported (p14II18-20) as well as a clear correlation between flashiness and basin area (p14II20-22 + Figure 10); these make perfect sense, but would they be different if flow-based flashiness indicators had been used? Please add a comparison to other flashiness indicators.

Reply: The reviewer is right, we will make sure to correct Table 2 to 3 in the text. We will add results for the most commonly used flashiness indicators, the R-B Index, and compare to our results for IAT-derived flashiness.

RC: In the very end of the section (p14ll30-34) a discussion of results not shown is given indicating that the manuscript would benefit from addition of a supplement containing results from all catchments and also the further analysis that has apparently been carried out.

Reply: The reviewer refers to this text in the manuscript "flashiness indexes computed for different sampling resolutions remain almost unchanged up to a transition range (8-16 days)." We will add the results to the supplement in the form of an additional figure.

RC: Multifractal analysis The whole section on scaling (p15II1-29) could really be shortened to one sentence simply stating that scaling is great for both flow and inter amounts accompanied by the left side of Figure 11 unless you can show that there is a statistical significant better fit of one of them. Also the identified departures from linearity (p15II6-88 and the right side of Figure 11) should be statistically significant to be relevant for discussion. Please provide relevant statistics to support the conclusions drawn or shorten the section.

Reply: We propose to keep this section on the results of multifractal analysis and

**HESSD**
articulate it around the two main aspects: (1) IATs scale better than flows (e.g., we will provide R2 values in the log-log plots) and (2) UM parameters C1 and alpha are different between the two approaches. Moreover, the C1 and alpha values for IATs are less sensitive to the selected range of scales.

RC: Conclusions In the conclusions it is stated that: "Flows sampled over fixed time intervals did not clearly exhibit this transition. This is result of peak flow variability being poorly sampled by fixed time window sampling." (p16ll28-29) but until you add results where you utilize the 15 minutes resolution this cannot be concluded.

Reply: this relates to the comment made for the Data section, where we explained the minimum resolution of analysis or minimum observable scale for IAT analysis is imposed by the definition of a maximum acceptable error.

RC: Another sentence: "Based on inter-amount times distribution we were able to define a flashiness indicator that incorporates both the rising and falling components of the hydrological response" (p17II28-29) seems to be unsupported as the flashiness indicator, as I have understood it, really only tell how many hours of mean flow one can expect as peak flow within a given much shorter time frame (e.g. an indicator of 100 hours for 15 minutes inter-amount times mean that the 99%-tile peak flow is 400 times the mean flow) and how the rise and fall of the peaks are incorporated is not clear.

Reply: the reviewer is right, this way of formulating the conclusion is not correct. We will rephrase the conclusion, make clear that the flashiness indicator is based on 1%-values and we will add a comparison to R-B index values.

RC: It is also concluded from the multifractal analysis that: "This showed that interamount times can help better predict peak flow characteristics at small unobservable scales based on coarse resolution data. Additionally, they provide new interesting alternatives for the stochastic modelling and downscaling of flow data." (p17II18-20) and "Scaling analysis showed that inter-amount times provide a promising way to better predict peak flow characteristics at small unobservable scales from coarse resolution HESSD
data" (p17ll31- 32) but this was not discussed at all before in the manuscript and if it is true you should really add results to support this.

Reply: We will make sure to make a clearer distinction in the text between observational resolution and sampling resolution for analysis. The point we want to make is that since IATs are based on fixed flow accumulation values, they will be sampled at higher than the average sampling resolution during peak flows and at low resolution during base flow. Furthermore, since scaling analysis showed better scaling behaviour, especially for higher order moments, downscaling to smaller scales based on IATs is likely to produce more robust results.

RC: Figures In general the figures need some work before publication.

RC: The fonts used are generally very small (e.g. the legend for Figure 1 which is unreadable when printed). Reply: we will make sure to increase the font

RC: The use of sub-figure numbering is inconsistent between text and figures (e.g. Figure 4c and d are not mentioned in the caption and for Figures 6, 8 and 9 the subplot labels are missing). Reply: we will make sure to correct figure references

RC: In Figure 9 there is no marking of which color corresponds to which data set. Reply: we add an explanation to the figure caption

RC: Inconsistent use of 'IATs' (Figure 11) and 'inter-amount time' (Figure 8) as well as 'Flow' (Figure 2) and 'Amounts' (Figure 11) and 'medcouple' (Figure 7) and 'MC' (Figure 5). Reply: we will make sure to correct the terms in captions and in figures

RC: What are the units of the x-axes of Figure 6? Reply: we will add the units in the caption

RC: Also put the unit directly on the x-axes of Figure 4 and not only in the text of the figure. Reply: we add unit to the x-axes

RC: For Figure 8 the x-axis seem confusing. For the inter-amount times the volumes
are based on time and should be reported something like "0.51 mm (12h)" but for the flow plot the axis should only be time. Similarly for Figure 11 where the x-axes for flow and inter-amount times should be different. Reply: we will show time only for the axes of the flow plots and show both time and normalised volumes for the axes of the IAT flows, in order to enable easier comparison between the two types of plots.

RC: Concluding remark Indeed, this approach is very interesting as it generate data sets with higher sampling frequency when high flow occur and lower sampling frequency for low flow periods. This is nicely pointed out by the authors. However, from the reported results I am not convinced that the methodology adds so much to the field.

Reply: The method presented here is proposed for analysis of existing datasets, the idea is to complement analysis based on conventional sampling by IAT sampling analysis. The paper shows what additional insights can be obtained, e.g.: flashiness, flow peaks missed at given observational resolution, statistical results less sensitive to outliers, future opportunities for up and downscaling. Additionally, the approach shows that adaptive sampling for collection of flow observations would be beneficial and in which cases most improvements can be expected.

RC: It is also unclear how exactly the authors see that this knowledge can be utilized in future research as 1) very high resolution flow data is used in this study and 2) it is unclear how the results can be used to better describe watersheds with much coarser data available as it is already pointed out that considerable uncertainty is associated with estimation of the peak flow from 15 minutes resolution data; how will than then look if only daily data is available?. Also downscaling of coarse flow data is mentioned, but for the same reasons as just mentioned it is very unpredictable how this will work.

Reply: For opportunities with respect to downscaling of flow data, we refer to previous replies related to the scaling analysis. Regarding the comment on observational resolution, the authors think it safe to say that nowadays automated gauges are replacing manual gauges in many places and with automated gauges observational resolutions
of 15 minutes are quite common and likely to go down to 5 min and 1 min resolution. Especially in urban areas, given the high flow variability at small scales.

References:

Baker, D. B., et al. A new flashiness index: Characteristics and applications to midwestern rivers and streams1. (2004): 503-522.

Schleiss, M., and J. A. Smith. Two Simple Metrics for Quantifying Rainfall Intermittency: The Burstiness and Memory of Interamount Times. Journal of Hydrometeorology 17.1 (2016): 421-436.

---

## Author Comment (AC2) · 6 Dec 2016

Review of "Characterising hydrological response in urban watersheds based on inter-amount time distributions" by ten Veldhuis and Schleiss. Referee #2

RC: Reviewer comment; Reply: authors' reply

RC: This paper applies an existing framework (Schleiss and Smith, 2016) for analysing inter-amount-times (IAT) to time series of daily streamflow from 17 urbanised watershed monitored by the USGS. The stated aim of the paper is to explore how IAT can be used to characterize the hydrological response and compare results to analysis based on traditional analysis techniques. In general I found the manuscript difficult to read

for a number of reasons; The introduction is very unstructured. It starts by discussing effects of urbanisation, then gives a general A-Z of general challenges in applied hydrology, incl. flood frequency analysis, flow duration curves, multivariate statistics, unit hydrographs, baseflow separation, before ending-up with a very detailed summary of scaling issues with a high level of assumed knowledge on behalf of the reader. I suggest a much more focussed introduction is necessary to better justify the scientific gaps being addressed by the study.

Reply: We agree with the reviewer's suggestion for a more focused literature review in the introduction and propose to make the following changes: (1) Frame the analysis more clearly as a statistical analysis and clearly state shortcomings of traditional statistical/frequency analysis approaches: separation between peak flow analysis, based on annual maxima or POT values versus low flow analysis, based on annual flow minima or 75-99% exceedance flows. Combining both in one analysis is difficult because flow distributions are highly skewed. (2) We will focus the literature review on 3 topics: Statistical analysis of flow time series - existing approaches for flood frequency analysis, low flow frequency analysis, benefits of being able to combine both aspects in a single framework, especially for analysing impacts of change in flow regimes such as for urbanisation impacts on hydrological response; Flashiness index (as requested by reviewer #1): review existing flashiness indices, esp. most frequently used R-B flashiness index (Baker, 2004); Scaling analysis: summarising literature on scaling relationships based on traditional flow time series; studies have shown scaling results to be dependent on the original resolution of the analysis (daily/hourly/5-min data aggregations show different scale breaks). More robust scaling relationships are needed to provide reliable results for up- and downscaling of flow data.

RC: The aim is presented very loosely as 'to explore how inter-amounts can be used to characterise hydrologic response for a range of (semi)urban watersheds in North Carolina, US.' I think it would be more convincing if the aim could be linked more strongly to a distinct problem/gap when using existing methods for analysing the hydrological

response from urban watersheds. What is the actual scientific and/or applied problem being addressed here?

Reply: we will rephrase the objective of the paper to make it more specific, addressing representation of high flows and low flows in frequency distributions (lower CVs for IAT, differences in skewness and medcouple values), flashiness characterisation, scaling behaviour

RC: I was struggling with some of the technical description in Section 2. This is partly down to a number of key places where the notation is hard to follow (see more detailed comments below), but also because I could not follow how this method was going to add new insight that was not available from a direct analysis of runoff time series. Again, I think a more focussed and readable introduction might have been helpful here.

Reply: We appreciate the reviewer's comment and understand that the notion of inter-amount times and what differences it implies for data sampling strategy and statistical representation of the data are hard to grasp at first view. We will add explanation on the expected benefits on IAT analysis, as explained above, we will rephrase the objective to provide better guidance for the reader.

RC: The results discussion (Section 3) is hard to follow. Maybe consider introducing more subsection and better explain how the different analysis comes together to answer the scientific aim, rather than at present where I get the impression it is a series of independent and individual analysis undertaken because it is possible.

Reply: We will follow the reviewer's suggestion and add subsection headings in Section 3 as well as make sure to reflect more clearly on how results answer to the objectives of the study.

RC: The main conclusion seems to be that flashiness is related to watershed area, but that no link to urbanisation could be identified. While this is, of course, not the fault of the researchers it does perhaps suggest that the title of the paper (urban watersheds)

is not really appropriate as no new insight into the effects of urbanisation seems to have been discovered.

Reply: Indeed, no direct link between urbanisation and IAT properties could be identified. However, we believe IATs still provide an interesting alternative framework for studying flow properties across scales. They come with many advantages, e.g., less variable distributions across scales and better scaling, and can therefore be of advantage in many statistical analyses. As such, the IAT approach is meant to complement traditional analysis techniques, not replace them. We will modify the text to better convey this message and put less weight on the urbanisation issue.

RC: I don't think the strong conclusion on page 17, line 21-22 is justified as it is not clear what characteristics is being referred to that could not have been established using conventional time series?

Reply: the motivation for this conclusion is provided in the sentences following lines 21-22: IAT analysis: patterns of low regulation could be identified more clearly, the loss of information on flow variability during high flows could be quantified and different aspects of flashiness were identified compared to traditional flashiness indicators. We will rephrase this section to summarise the arguments more distinctly, with more explicit links to the presented results.

RC: Detailed comments:

Reply: we will review notations and definitions and make sure to correct and add additional explanation as indicated by the reviewer

RC: I think HESS uses British rather than US spelling, so catchment rather than watershed Page 2, line 25: Define 'scale-break' Page 3, line 5: Not sure what 'moments q[0.1-4]' signifies? Eq. (4): this equation sums over qi, but in the lines above the sample flow time series is defined in terms of r (line 14). Should it be 'r' in Eq(4) or else 'qi' need to be defined somewhere Eq (5): I don't understand the notation used in this

equation. What does 'Qt,nq-nq' mean? Eq (6): I don't know how to link this equation to Eq. (5) – I think some more explanation is required here. Eq(7): 'T' is not defined anywhere? Page 6, lines 23-24: Given that the flashiness is one of the main conclusions of the study, I think a more comprehensive description of the concept is required, for example include a conceptual figure. Page 7, line 16: What is 'Scott's rule'? Page 19=0, line 14: what does 'cq' refers to?

References:

Baker, D. B., et al. A new flashiness index: Characteristics and applications to midwestern rivers and streams1. (2004): 503-522.

Schleiss, M. and J. A. Smith. Two Simple Metrics for Quantifying Rainfall Intermittency: The Burstiness and Memory of Interamount Times. Journal of Hydrometeorology 17.1 (2016): 421-436.

---

## Author Response (AR1)

Dear Editor,

Based on the comments we received from the 2 referees who reviewed the first version of the manuscript, we have revisited several parts of our analysis and thoroughly revised the manuscript. In the "track-changes" version of the manuscript, we have indicated the sections where major changes were made (text removed and new texts added). We have included additional results on computation of R-B flashiness indices and compared these to the IAT flashiness index we have defined. Below we briefly explain how each of the referees' comments was dealt with in the revision. We thank the two reviewers for their critical and constructive comments which have helped us to deepen our analysis and bring out more clearly some interesting findings based on adaptive sampling using inter-amount times.

Review of "Characterising hydrological response in urban watersheds based on inter-amount time distributions" by ten Veldhuis and Schleiss
Referee #1

The study propose a methodology for sampling flow data sets from water courses in urban watersheds based on samples of equal volume.

Data
The stream gauge data used in this study has a temporal resolution of maximum 15 minutes (p4l17) but all results reported are for longer aggregation periods: in Figure 8 minimum of 12 hours and in Figure 9 and 10 a minimum of 3 hours. With data available at 15 minutes resolution it should be possible to detect rapid changes in the flow and the manuscript seriously lack comparisons at the sub-hourly scales.
*Reply: The minimum scale at which analyses have been conducted was determined by the definition of minimum inter-amount scale, as explained in section 2.4, equations (8) and (9). To limit estimation errors of inter-amount times, we set the mean of absolute relative errors to be smaller than 50%. This requirement determines the minimum sampling scale for IAT analysis; flow analysis and IAT were conducted over the same range of scales. We would like to emphasise that because IAT sampling is an adaptive sampling strategy, much shorter IATs than the mean sampling scale are observed during periods of high flows. We revised sentences explaining the equations (8) and (9) to make this point more clear (p.8, l17-22). Additionally, we added a paragraph at the start of section 3 to explain that analyses were first conducted at 24 hour scale, then across a range of scales (p12, l20-23).*

One exception is in Table 3 where the flashiness index is reported for 15 min observational resolution; but here it is unclear whether the 99%-tile of the flow measurements at 15 minutes resolution would give the same answer. Please investigate this and add the results to the discussion.
*Reply: We added a paragraph analysing 10-90% and 1-99% ranges for both flows and IATs (section 3.3, p17-18, l20-23,10-14). We also expanded the section on flashiness index by adding results for computation of Richards-Baker flashiness indices, a flashiness index commonly used in the hydrology literature. We compared results for R-B flashiness indices with those for IAT flashiness based on the 99% of IATs (sections 2.7 and 3.4).*

As illustrated by Figure 3 the inter-amount time methodology result in much less data points than the original data set, but it is very unclear why it provides a better data basis for discussing the hydrological properties than the original 15 minute resolution data.
*Reply: This is a misunderstanding ; we have rephrased the captions for figures 3a and 3b to explain that figure 3a shows original observed time series and figure 3b compares flow data and inter-amount data at a same resolution to illustrate how sampling strategies differ.*

"Missing data were treated as zeros" (p4l23): how does this influence the results and the estimation error?
*Reply: We have added several lines to discuss the effect of missing data on IATs (p7 l3-9)*

The catchments used in the study are to some degree sub catchments of each other. This could mean that some stream gauges are correlated (e.g. 507 and 530 (Figure 1)) but this is not discussed in the manuscript. This should be discussed in general and specifically in relation to the results where all catchments are discussed (e.g. Tables 2 and 3 and Figures 5, 7, 20 and 13).
*Reply: We have added a discussion on relations between results for catchments that are subcatchments of each other in section 3.2 (p15, l9-15) and section 3.3 (p16, 20-21) and added a table (Table 4) summarising results for the 3 sets of related subcatchments in our study.*

Figure 1 is very hard to read and could benefit from being enlarged. Also, some will not know where in the world Charlotte, NC, is and it would be beneficial to add a panel of the North American East Coast with a marking of where the study area is.
*Reply: We have added a panel showing the location of Charlotte and increased font size.*

Figure 2 is not providing any insight and should be removed.
*Reply: Figure 2 visualises times series of flows and associated inter-amount times and shows how the inter-amount times have lower variability and fewer outliers as a result of the different sampling strategies. These aspects are discussed later in relation to the respective statistical distributions. Since many readers will never have seen IAT time series we prefer to keep this figure.*

Inter-amount times
The definition of inter-amounts (p4l24-p5l2) is brief and to the point. The section on normalization of inter-amounts (p5ll3-12) is also brief and the arguments for the methodology are good. The section on sample estimate (p5l13-p6l18) is somewhat harder to follow. The section thoroughly explain how to convert a time series of flow measurements to a series of inter-amount times and the possible error introduced by the approach but in the results section a measure of the error associated with the present 15 minute resolution data and the present catchments is not reported. This is really needed as it should be really limited how important this is at this fine temporal resolution of the flow data.
*Reply: The error in flow measurement associated with temporal observational resolution, in relation to the real flow variability, is not known for these datasets. For this, a set of observations at higher temporal resolution would be required, for comparison. As reported in the paper, flow data are derived from water depth measurements using flow-rating curves that regularly recalibrated by USGS. Documentation reporting quantitative error estimates is not available to the authors.*

CV, skewness and medcouple are used to compare inter-amount time and flow distributions. In general the discussion of the results (p9l5-p11l9) are for daily values. This section could be much more interesting by adding results for higher resolution since the native data resolution is so much higher than the daily scale.
*Reply: As explained above, the minimum scale for IAT analysis is limited by our definition of acceptable error in IAT estimation (equations (8) and (9)). Minimum scales vary between 3 and 13.75 hours (see table 5). We first analysed and compared IAT analysis at daily scale, sections 3.1-3.3, then then over the range of available scales, in sections 3.4-3.6. A brief introduction has been added at the start of section 3, to explain this (p12, l20-23).*

At p10ll28-29 a bi-modal histogram for catchments with low flow regulation is discussed but not shown, please add these in a supplement.
*Reply: A supplement has been added, presenting histograms of flows and IATs for all gauges as well as histograms of differences in flows and IATs for all gauges and quantile plots for all gauges.*

From p10l30 to the end of the section is repetition that could and should be left out of the manuscript.
*Reply: We have rephrased and shortened this paragraph (p14, l9-13)*

The distribution of changes in inter-amount times is used to identify rapidly increasing and decreasing trends (p11ll10-30). Figure 6 summarizes the results (again at daily scale) between flow based and inter-amount time based investigations but I cannot see how there can be both inter-amount times and flows in the figure. And it is not clear how to quantitatively get more knowledge from the inter-amount times since the qualitative conclusions will be the same between flows and inter-amount times (even though the skew will be in opposite directions). Please elaborate on this and correct the figure.
*Reply: Figure 6 has been changed, the new version presents histograms for differences in flows and in IATs for 2 gauges (histograms for other gauges available in the supplement). We have revisited the analysis of differences in IATs and substantially revised the text section 3.2 to present clear conclusions of this analysis.*

Inter-amount times are further compared to flows in Figure 8. Figure 8 is a really good example of all the problems you get from having box-plots on a log scale. For both values span several orders of magnitude and is vastly skewed (as indicated by the large difference between the mean and the median). The associated discussion (p11l31-13l15) is very hard to follow and whether a given percentile is following a power law (p12ll10-12) or not is effectively impossible to see from the figure. I would suggest a form of normalization of the results to avoid the logarithmic axes in Figure 8 and make the scaling discussion much more accessible.
*Reply: The reviewer is correct in that the values span 4 orders of magnitude in scale, hence our choice for visualising the Q-Q plots on a log-scale. Normalisation as suggested by the reviewer has been applied, the current figures are based on values normalised by basin area. The distributions are indeed vastly skewed; understanding the origins of the skewness and how it differs between flows and IATs, between gauges and how it changes with scale is what we are analysis in this this subsection (3.3). The text in section 3.3 has been thoroughly revised to improve the explanations and discussion.*

Flashiness
From the very first sentences of the abstract (p1ll1-5) flashiness is highlighted as a key parameter where inter-amount time distributions can really make a difference. In the introduction it is concluded from literature that it is difficult to predict the flashiness of urban watersheds, but no methodologies or results are directly presented. Please add a more thorough introduction on how flashiness is normally calculated from traditional flow data.
*Reply: We have added a brief literature review on flashiness indicators to the introduction (section 1.2, p3, l4-18). The most commonly used is the R-B flashiness index, introduced by Baker (2004):*
*R-B Index* $= R - B \ Index = \frac{\sum_{i=1}^{n} |q_i - q_{i-1}|}{\sum_{i=1}^{n} q_i}$
*We have computed this index for our datasets and compared results with those for the flashiness indicator derived from IAT analysis. Results are presented in a largely new section, section 3.5.*

The flashiness indicator formulated in the study (p6ll23-24) is very briefly described and a discussion of why this choice was made and why this is a good indicator for flashiness is completely lacking. Please add these.
*Reply: We have added a new section 2.7, explaining the definition of IAT flashiness and that of R-B flashiness (p.11)*

The results and discussion for flashiness and minimum observable scale (p13l32 +p14) is not easy to follow. In the first section (p13l33-p14l10) It is discussed that very high peak flows cannot be measured correctly every time with 15 minutes resolution data; but is this interesting at all and is it important to know flow variation at this high resolution? Please add a discussion of this.
*Reply: We have thoroughly revised this section and added discussion and comparison with R-B flashiness index (section 3.5, p20-21).*

In the next section ('Table 2' should really be 'Table 3') a clear correlation between minimum observable scale and flashiness is reported (p14ll18-20) as well as a clear correlation between flashiness and basin area (p14ll20-22 + Figure 10); these make perfect sense, but would they be different if flow-based flashiness indicators had been used? Please add a comparison to other flashiness indicators.
*Reply: We have added values for R-B flashiness in the table 5 and corrected table references in the text. Results are discussed in a new section on flashiness, section 3.5.*

In the very end of the section (p14ll30-34) a discussion of results not shown is given indicating that the manuscript would benefit from addition of a supplement containing results from all catchments and also the further analysis that has apparently been carried out.
*Reply: We have looked into this more thoroughly and found that investigating variation of flashiness with scale is not straightforward. As discussed in section 3.5, the well-established R-B flashiness index does not allow comparions between values computed at different scales. IAT flashiness can be computed and compared at different scales, but this adds a lot of complexity to the analysis due to the interaction between data resolution and IAT sampling scale. Results require thorough interpretation and discussion, which we plan to work on in the future, but is beyond the scope of the current paper.*

Multifractal analysis
The whole section on scaling (p15ll1-29) could really be shortened to one sentence simply stating that scaling is great for both flow and inter amounts accompanied by the left side of Figure 11 unless you can show that there is a statistical significant better fit of one of them. Also the identified departures from linearity (p15ll6-88 and the right side of Figure 11) should be statistically significant to be relevant for discussion. Please provide relevant statistics to support the conclusions drawn or shorten the section.
*Reply: We have revised parts of this section to better explain the two main aspects of multifractal analysis of IATs and flows: (1) IATs scale better than flows (we have provide R2 values in the log-log plots and in table 5) and (2) UM parameters C1 and alpha are different between the two approaches. Moreover, the C1 and alpha values for IATs are less sensitive to the selected range of scales.*

Conclusions
In the conclusions it is stated that: "Flows sampled over fixed time intervals did not clearly exhibit this transition. This is result of peak flow variability being poorly sampled by fixed time window sampling." (p16ll28-29) but until you add results where you utilize the 15 minutes resolution this cannot be concluded.
*Reply: We have revised the text in the related section (section 3.4, p17, l15-18) as well as in the conclusions (p25, l10-15) to make this point more clear.*

Another sentence: "Based on inter-amount times distribution we were able to define a flashiness indicator that incorporates both the rising and falling components of the hydrological response" (p17ll28-29) seems to be unsupported as the flashiness indicator, as I have understood it, really only tell how many hours of mean flow one can expect as peak flow within a given much shorter time frame (e.g. an indicator of 100 hours for 15 minutes inter-amount times mean that the 99%-tile peak flow is 400 times the mean flow) and how the rise and fall of the peaks are incorporated is not clear.

*Reply: We have replaced this text by a new conclusion based on our more extensive analysis of IAT flashiness and new results for R-B indices (p25-26, l29-30, l1-9)*

It is also concluded from the multifractal analysis that: "This showed that inter-amount times can help better predict peak flow characteristics at small unobservable scales based on coarse resolution data. Additionally, they provide new interesting alternatives for the stochastic modelling and downscaling of flow data." (p17ll18-20) and "Scaling analysis showed that inter-amount times provide a promising way to better predict peak flow characteristics at small unobservable scales from coarse resolution data" (p17ll31- 32) but this was not discussed at all before in the manuscript and if it is true you should really add results to support this.
*Reply: We have replaced this text by a new conclusion (p26).*

Figures
In general the figures need some work before publication.

The fonts used are generally very small (e.g. the legend for Figure 1 which is unreadable when printed).
*Reply: we have increased the font size of some of the figures*

The use of sub-figure numbering is inconsistent between text and figures (e.g. Figure 4c and d are not mentioned in the caption and for Figures 6, 8 and 9 the sub-plot labels are missing).
*Reply: we have corrected figure references*

In Figure 9 there is no marking of which color corresponds to which data set.
*Reply: we have added an explanation to the figure caption*

Inconsistent use of 'IATs' (Figure 11) and 'inter-amount time' (Figure 8) as well as 'Flow' (Figure 2) and 'Amounts' (Figure 11) and 'medcouple' (Figure 7) and 'MC' (Figure 5).
*Reply: we have corrected the terms in captions and in figures*

What are the units of the x-axes of Figure 6?
*Reply: we have added the units in the caption*

Also put the unit directly on the x-axes of Figure 4 and not only in the text of the figure.
*Reply: we have added unit to the x-axes*

For Figure 8 the x-axis seem confusing. For the inter-amount times the volumes are based on time and should be reported something like "0.51 mm (12h)" but for the flow plot the axis should only be time. Similarly for Figure 11 where the x-axes for flow and inter-amount times should be different.
*Reply: we have changed the of units of the axes in the plots*

Concluding remark
Indeed, this approach is very interesting as it generate data sets with higher sampling frequency when high flow occur and lower sampling frequency for low flow periods. This is nicely pointed out by the authors. However, from the reported results I am not convinced that the methodology adds so much to the field.
*Reply: We have thoroughly revised large parts of the results section and extended our analysis of flashiness. We believe that the results that are presented in the revised manuscript bring out much more clearly the additional insights that were obtained from IAT analysis.*

It is also unclear how exactly the authors see that this knowledge can be utilized in future research as 1) very high resolution flow data is used in this study and 2) it is unclear how the results can be used to better describe watersheds with much coarser data available as it is already pointed out that considerable uncertainty is associated with estimation of the peak flow from 15 minutes resolution data; how will than then look if only daily data is available?. Also downscaling of coarse flow data is mentioned, but for the same reasons as just mentioned it is very unpredictable how this will work.

*Reply: For opportunities with respect to downscaling of flow data, we refer to previous replies related to the scaling analysis. Regarding the comment on observational resolution, the authors think it safe to say that nowadays automated gauges are replacing manual gauges in many places and with automated gauges observational resolutions of 15 minutes are quite common and likely to go down to 5 min and 1 min resolution. Especially in urban areas, given the high flow variability at small scales.*

Review of "Characterising hydrological response in urban watersheds based on inter-amount time distributions" by ten Veldhuis and Schleiss
Referee #2

This paper applies an existing framework (Schleiss and Smith, 2016) for analysing inter-amount-times (IAT) to time series of daily streamflow from 17 urbanised watershed monitored by the USGS. The stated aim of the paper is to explore how IAT can be used to characterize the hydrological response and compare results to analysis based on traditional analysis techniques.

In general I found the manuscript difficult to read for a number of reasons; The introduction is very unstructured. It starts by discussing effects of urbanisation, then gives a general A-Z of general challenges in applied hydrology, incl. flood frequency analysis, flow duration curves, multivariate statistics, unit hydrographs, baseflow separation, before ending-up with a very detailed summary of scaling issues with a high level of assumed knowledge on behalf of the reader. I suggest a much more focussed introduction is necessary to better justify the scientific gaps being addressed by the study.

*Reply: We have thoroughly revised the introduction section, following the reviewer's suggestion for a more focused literature review. We have also added a brief literature review on flashiness indices*

The aim is presented very loosely as 'to explore how inter-amounts can be used to characterise hydrologic response for a range of (semi)urban watersheds in North Carolina, US.' I think it would be more convincing if the aim could be linked more strongly to a distinct problem/gap when using existing methods for analysing the hydrological response from urban watersheds. What is the actual scientific and/or applied problem being addressed here?

*Reply: We have rephrased the scope and objective of the paper in section 1.4 in the introduction section (p 5).*

I was struggling with some of the technical description in Section 2. This is partly down to a number of key places where the notation is hard to follow (see more detailed comments below), but also because I could not follow how this method was going to add new insight that was not available from a direct analysis of runoff time series. Again, I think a more focussed and readable introduction might have been helpful here.

*Reply: Parts of the text in section 2 have been revised, equations in section 2 have been checked and corrected in case they were not consistent and, as explained above, we revised the introduction section to make the scope and objective of the paper more clear.*

The results discussion (Section 3) is hard to follow. Maybe consider introducing more subsection and better explain how the different analysis comes together to answer the scientific aim, rather than at present where I get the impression it is a series of independent and individual analysis undertaken because it is possible.

*Reply: The results section has been thoroughly revised, as indicated in the revised, tracked-changes manuscript. We have added paragraph breaks to better separate topics discussed in the text.*

The main conclusion seems to be that flashiness is related to watershed area, but that no link to urbanisation could be identified. While this is, of course, not the fault of the researchers it does perhaps suggest that the title of the paper (urban watersheds) is not really appropriate as no new insight into the effects of urbanisation seems to have been discovered.

*Reply: In the revised text of chapter 3 and in the revised conclusions presented in chapter 4, the effect of urbanisation on results we obtained in IAT analysis are discussed more clearly and explicitly.*

I don't think the strong conclusion on page 17, line 21-22 is justified as it is not clear what characteristics is being referred to that could not have been established using conventional time series?

*Reply: We have revised this section to summarise the arguments more distinctly, with more explicit links to the presented results.*

Detailed comments:
I think HESS uses British rather than US spelling, so catchment rather than watershed
*Reply: We have replaced watershed by catchment throughout the text.*

Page 2, line 25: Define 'scale-break'
Reply: this text has been revised
Page 3, line 5: Not sure what 'moments q[0.1-4]' signifies?
Reply: the text has been revised and the definition of moments is explained better.
Eq. (4): this equation sums over qi, but in the lines above the sample flow time series is defined in terms of r (line 14). Should it be 'r' in Eq(4) or else 'qi' need to be defined somewhere
Reply: the equation has been corrected
Eq (5): I don't understand the notation used in this equation. What does 'Qt,nq-nq' mean?
Reply: the equation has been corrected
Eq (6): I don't know how to link this equation to Eq. (5) – I think some more explanation is required here.
Reply : the equations have been corrected
Eq(7): 'T' is not defined anywhere?
Reply : the equations has been changed, T is now defined already in equation (3)
Page 6, lines 23-24: Given that the flashiness is one of the main conclusions of the study, I think a more comprehensive description of the concept is required, for example include a conceptual figure.
reply: a new section on flashiness indices has been added in the methods section (2.7) and in the results section (3.5)
Page 7, line 16: What is 'Scott's rule'?
reply: Scott's rule defines how bin widths are determined for histograms. A reference was provided
Page 19=0, line 14: what does 'cq' refers to?
reply: abbreviation 'cq' has been removed.

**Statistical analysis of hydrological response in urbanising catchments based on adaptive sampling using inter-amount times**

Marie-claire ten Veldhuis[1,3] and Marc Schleiss[2,3]

[1]Delft University of Technology, Watermanagement Department
[2]Delft University of Technology, Geosciences and Remote Sensing Department
[3]Princeton University, Hydrometeorology Group

*Correspondence to:* Marie-claire ten Veldhuis (j.a.e.tenveldhuis@tudelft.nl)

**Abstract.** Urban catchments are typically characterised by a more flashy nature of the hydrological response compared to natural catchments. Predicting flow changes associated with urbanisation is not straightforward, as they are influenced by interactions between impervious cover, basin size, drainage connectivity and stormwater management infrastructure. In this study, we present an alternative approach to statistical analysis of hydrological response variability and basin flashiness, based on the distribution of inter-amount times. We analyse inter-amount time distributions of high-resolution streamflow time series for 17 (semi)urbanised basins in North Carolina, US, ranging from 13 km$^2$ to 238 km$^2$ in size. We show that in the inter-amount times framework, sampling frequency is tuned to the local variability of the flow pattern, resulting in a different representation and weighting of high and low flow periods. [c1]This leads to important differences in the way the quantiles, mean, coefficient of variation and skewness of the distributions vary across scales and results in lower mean intermittency and improved scaling. Moreover, we show that inter-amount times distributions can be used to detect regulation effects on flow patterns, identify critical sampling scales and characterise flashiness of hydrological response. The possibility to use both the classical approach and the inter-amount time framework to identify minimum observable scales and analyse flow data opens up interesting areas for future research.

**1 Introduction**

Hydrological response in urban [c2] [c3]catchments tends to be more flashy compared to natural ones as a result of their higher degree of imperviousness. Increase in flashiness is typically characterised by shorter response times to rainfall, higher runoff ratios and higher peak flows (Berne et al., 2004; Smith et al., 2005). On the other hand, high impervious degrees may [c4]reduce base flows and [c5]lead to intermittent flow during dry periods. At the same time, urbanisation is usually tied to development of urban drainage infrastructure, [c6]associated with artificial flow control as well as higher peak flows due to increased drainage
* * *
[c1] *Text added.*
[c2]
[c3] *Text added.*
[c4] *Text added.*
[c5]
[c6] *Text added.*

20 connectivity. Predicting the degree of flashiness or base flow reduction associated with urbanisation is not straightforward, as it depends on the interplay of impervious cover, basin size and shape, soil properties, basin slope, drainage connectivity and control structures such as detention ponds, weirs and pumps (Emmanuel et al., 2012; Fletcher et al., 2013; Smith et al., 2013). [c7]Traditional analyses of flow time series tend to focus on specific aspects and flow characteristics, aiming for example at predicting low flow durations or peak flow magnitudes. For analysis of change in hydrological response, it may be beneficial

5 to combine both peak flow and low flow statistics into a single framework. This applies in particular to the context of urban hydrology where urbanisation and human intervention alter both high flow and low flow characteristics of the hydrological response. Combining both aspects in a single analysis is difficult, as flow distributions are highly skewed and frequencies of low and high flow values are very different. In this paper, we show how alternative sampling of flow time series based on inter-amount times leads to more balanced statistical distributions, better representation of both high and low flows in a single

10 framework and more robust behaviour of statistical distributions across scales.

**1.1  Statistical analysis of hydrological response**

Many authors have investigated methods for characterising hydrological response and changes therein, including [c1]univariate analysis and multivariate statistics, combining several hydrograph properties such as flood peak, flood volume and flood duration (e.g., Salvadori and De Michele, 2004; Favre et al., 2004; Grimaldi and Serinaldi, 2006; Vittal et al., 2015). [c2]Traditional

15 statistical analysis techniques tend to focus on either left or right tail properties of statistical distributions, but not necessarily using the same statistical framework. Low flow analyses for example are primarily concerned with the total time the flow stays below a critical threshold (see e.g. Smakhtin (2001) for an extensive review). [c3]By contrast, peak flow analysis puts more weight on total accumulated flows at a given time scale using annual flow maxima or peak-over-threshold values to derive extreme value statistics and establish flood frequency curves (e.g., Stedinger, 1983; Lang et al., 1999; Villarini et al., 2009;

20 Smith and Smith, 2015). [c4]Both approaches are valid and solidly rooted in the context of extreme event analysis with numerous applications in drought and flood risk analysis. However, the statistical frameworks they rely on are not necessarily the same. Low flow analysis favours 'time' as a random variable. Peak flow analysis on the other hand treats the 'flow amount' over a fixed time interval as the main random quantity. This might seem more intuitive to many but there is no strong compelling reason to prefer one approach over the other a priori. For example, one might as well adopt an alternative framework in which

25 the unknown random variable is the 'time' necessary to cumulate a fixed, critical amount of flow. This approach is known as the inter-amount time (IAT) method (Schleiss and Smith, 2016) and has been previously proposed to analyse the properties of intermittent rainfall time series.[c5]One of the main goals of this paper is to apply the IAT formalism to flow time series to derive
* * *
[c7] *Text added.*
[c1] *Text added.*
[c2] *Text added.*
[c3] *Text added.*
[c4] *Text added.*
[c5] *Text added.*

properties of statistical distributions and compare the results to the ones obtained using the classical fixed-time framework. [c6] [c7]

**1.2 Change in hydrological response, basin flashiness**

[c1] An important characteristic that has been used to analyse change in hydrological response is basin flashiness, qualitatively
described by Poff (2002) [c2] as one of the indicators characterising change in natural flow regimes and how this affects the eco-
logical integrity of river ecosystems. Richter (1996) [c3] developed a set of 33 indices, the Indicators of Hydrological Alteration
(IHA), including indicators for conditions associated with flashiness, such as frequency and duration of high and low pulses,
and rate and frequency of change in flow conditions. Smith and Smith (2015) [c4] quantified flashiness of 5436 catchments in
the contiguous United States based on peak flows exceeding $1 \text{ m}^3\text{s}^{-1}\text{km}^{-2}$ normalised flows (i.e., flows normalised by basin
area). A frequently used index in the literature is the Richards-Baker (R-B) Flashiness Index (Baker et al., 2004), [c5] based
on the Richards pathlength (Gustafson et al., 2004). [c6] The R-B index is defined as the sum of absolute values of changes in
flow values divided by the total cumulative flow, and is usually computed at the daily time scale. Similar to the coefficient of
variation, it measures the relative dispersion of the flow at a given scale. A downside of the R-B index is that it highly sensitive
to the scale of analysis. Baker et al. (2004) [c7] argued that for smaller basins ($< 50 \text{ km}^2$) the use of hourly instead of daily flow
data should be considered to compute R-B flashiness index, but also found that R-B flashiness values computed at hourly scale
are highly sensitive to diurnal or other sub-daily low flow fluctuations. An important still unanswered question remains how to
overcome scale sensitivity of flashiness indicators in different hydrological basins. This is crucial for establishing how urbani-
sation impacts flashiness and how changes relate to basin characteristics such as size, slope, imperviousness degree and whether

[c6] ~~statistical approaches as well as hydrograph analysis and multivariate regression. Most of these studies have used daily flow time series, while more recently hourly and sub-hourly flow data series have increasingly been used. Traditionally, flow duration curves, representing the frequency distribution of flows, have been used to characterise hydrological response of a given basin. Wood and Hebson (1986) developed dimensionless flood frequency curves and investigated relations with basin and rainfall characteristics, length ratio, a geoclimatic scaling factor and dimensionless mean storm duration. Others have investigated the use of regional flood frequency curves to predict flood response in ungauged basins. Similarly, low flow frequency curves have been used to characterise the influence of prolonged periods of drought, see for instance Smakhtin (2001) for an extensive review. More recently, authors have concluded that univariate flood frequency curves are insufficient to describe flow response and multivariate approaches have been developed, combining several hydrograph properties such as flood peak, flood volume and flood duration~~

[c7] ~~Other approaches for characterising basin flow response and flood probability include establishment of Instantaneous Unit Hydrographs and design flood hydrographs. At the other end of the spectrum, base flow indices and subflow separation techniques have been developed to characterise low flow conditions. Others have investigated whether statistical properties derived from flow time series are still valid under changing climate conditions. One of the problems in analysing hydrological response across different events and basins is that hydrological response variables need to be normalised for comparison. This is usually done by dividing instantaneous flows and cumulative flow volumes by basin area. The downside of this approach is that definition of basin boundaries is prone to errors and that, especially for basins with inhomogeneous urbanisation coverage, basin area may imperfectly represent flow generation.~~

[c1] *Text added.*
[c2] *Text added.*
[c3] *Text added.*
[c4] *Text added.*
[c5] *Text added.*
[c6] *Text added.*
[c7] *Text added.*

urbanisation thresholds can be identified above which basin response is characteristically urban (Praskievicz and Chang, 2009).

c1

**1.3 Scaling analysis of hydrological flows**

[c2]Scaling behaviour of river flows has been investigated by various authors, aiming to identify characteristics length and time scales and to detect scale dependence of hydrological response processes. Among the various statistical methods that have been proposed to investigate scaling, fractals and multifractals are among the most popular and powerful. Approaches for fractal analysis include: spectral analysis based on $2^{nd}$ order properties, and trace moments analysis based on a wider range of statistical moments, typically between 0.1 and 4. The universal multifractal framework is based on the identification of scaling exponents summarising the changes in flow distributions across a given range of scales[c3], (see Schertzer and Lovejoy (1987) and Schertzer and Lovejoy (2011) for a review).

[revised manuscript text omitted]

[c2] *Text added.*

[c3] derived from time series analysis over a range of scales and a range of statistical moments

and only over a limited range of scales. Many studies report the existence of "scale breaks" at which scaling parameters change and significant departures from (multi)fractality can be observed. [c4] Table 1 [c5]summarises findings from selected scaling analyses of flow time series in the literature. It shows that the number and location of the scale breaks as well as the values of the multifractal parameters are sensitive to the method applied to estimate them and the resolution of the data used to conduct the

5   analysis. [c6]For example, Labat et al. (2013) [c7]performed spectral analysis and trace moments analysis for 30-minute flow times series and identified different flow regimes with scale breaks at 1 day for spectral and 16 hours for trace moments analysis. But when they performed the same analysis at daily and at 3 minute resolution, they identified different scaling regimes, with scale breaks at 16 days and 1 hour for daily and 3 minute resolution. Similarly, Sauquet et al. (2008) [c8]found different scaling regimes in their scaling analysis of flows for 34 basins, with scale breaks at 12 days for daily resolution and scale breaks varying

10  between 8.7 hours and 7 days across basins when using hourly data resolution, based on spectral analysis. When they applied trace moments analysis for the same time series at hourly resolution, they found no scale breaks for the lower order moments and scale breaks between 10 and 150 hours for higher order moments. This shows that while most flows exhibit some sort of scaling behaviour, the identified scaling laws are not very robust nor consistent, as they are dependent on analysis methods and data resolution.

**1.4   Statistical analysis of hydrological response based on adaptive sampling using inter-amount times**

[c1]In this paper, the IAT formalism is applied to flow time series and statistical distributions and scaling properties are compared to the ones obtained using the classical fixed-time framework. To do this, we use flow observations collected in 17 hydrological basins in Charlotte, North Carolina. We aim to investigate what effects an adaptive sampling strategy such as IAT sampling has

20  on statistical properties of the time series, in particular on the tails of the statistical distributions associated with peak flow and low flow extremes. [c2]The main problem with a fixed sampling rate, as in traditional flow time series analysis, is that it can only accurately represent frequencies of variations at time scales larger than a certain threshold. When frequencies higher than that exist, errors are introduced as information about the higher frequency variability is lost (Dippe and Wold, 1985). [c3]Increasing the sampling resolutions solves this problem, but results in oversampling of base flow values with respect to peak flows. An

25  alternative consists in adopting an adaptive sampling strategy, i.e., one that adapts the sampling rate to the variability of the signal itself (e.g., Feizi et al., 2011). [c4]This makes sense for processes that are very unevenly distributed in time (such as rainfall and hydrological flows), and means taking more samples during periods of high activity (e.g., peak flows following storm
* * *
[c4]

[c5] *Text added.*

[c6] *Text added.*

[c7] *Text added.*

[c8] *Text added.*

[c1] *Text added.*

[c2] *Text added.*

[c3] *Text added.*

[c4] *Text added.*

[revised manuscript text omitted]

---

## Author Response (AR2)

Review of "Characterising hydrological response in urban watersheds based on inter-amount time distributions" by ten Veldhuis and Schleiss

Referee #1
RC: Generally the revised manuscript appear much more coherent and the depth of the methodological explanations much more accessible. The inclusion of the Richards-Baker flashiness index for comparison greatly enhance the discussion and help emphasize where this framework is better and where traditional flow based sampling is better.

RC: The authors have responded well to almost all comments raised except for the concerns about scaling. In the conclusions it is stated that: (p11ll27-29) "Flows exhibited departures from multifractality for most basins, while IATs systematically scaled better than flows and showed departures from multifractality only for three basins subject to low flow regulation.", but from the discussion part it is clear that the $R^2$s for both flows and AITs are all over 0.99 under all conditions which in my book is excellent for natural systems. Please discuss and document it if the AIT scaling is significantly better than the flow scaling or loosen the conclusions to state that the scaling is generally good for both conditions.
AR: The reviewer is right in that both approaches exhibit relatively good scaling, in the sense of R2 values, even the ones with R2 closer to 0.990. It's good to note that goodness-of-fit usually reported for multifractal analyses are above 0.95 or 0.99. Still, the fit for IATs is better than for flows as shown by higher R2 values, especially for higher order moments.
We have rephrased the conclusion as follows (page 22, ll 15-18): "Both approaches exhibited relatively good scaling, as indicated by $R^2$ values above 0.99. IATs systematically scaled better than flows and showed departures from multifractality only for three basins subject to low flow regulation, while flows exhibited departures from multifractality for most basins."

Further, it is unclear whether the $R^2$ reported in the left side of Figure 11 is with or without consideration of scale breaks and, if it is without scale breaks, the significance of the presence of scale breaks has to be discussed as well.
AR: The $R^2$ values are without scale breaks, we have clarified this in the figure caption.
Exact testing of the significance of "scale breaks" is difficult. We performed a Davies test (Davies, 2002); results showed that for flows (q=2.4), two breakpoints were significant (p-value 0.001). For IATs (q=2.4), there was at least 1 significant breakpoint, but the test for 2 breakpoints returned a p-value of 0.071. This shows that scaling is slightly better for IATs than for flows. We have added this clarification in the text (page 18, ll 27-29): "Based on an approximate Davies test (Davies, 1987), two breakpoints were significant for flows (p-value 0.001). For IATs, there was at least 1 significant breakpoint, but the test for 2 breakpoints returned a p-value of 0.071. This shows that scaling is slightly better for IATs than for flows."

Reference:
Davies, R.B. (2002) Hypothesis testing when a nuisance parameter is present only under the alternative: linear model case. Biometrika 89, 484-489.

Referee #2
RC: The authors have made a comprehensive revision of the initial submission resulting in a much more readable manuscript. As such I think the manuscript is close to a stage where it can be published, but I still have a number of issues/questions/suggestions:

RC: Page 2, line 25: There might be a strong practical reason why engineers prefer to focus on peak flow rather than time, as flow magnitude determines hydraulic design?

AR: As we show later in the results chapter, both statistical distributions of flows and IATs can be used to derive information on return periods of peak flow, typically of interest for design and engineering. The point we want to make is that in IAT analysis, statistical properties of low flows and peak flows can be analysed using the same statistical framework. We have added the following sentence for clarification (page 1, ll 21-22):
"By doing so, both low flows and peak flows can be analysed using the same statistical framework."

RC: Page 7, top. Maybe spell out that the hard work of collecting and quality controlling the data was done by USGS staff
AR: Thanks for pointing this out. We have added the following sentence (page 5, ll 13-15):
"These curves were established based on protocols developed by USGS and include manual flow measurements during site visits performed by USGS staff. As part of this procedure, stage-discharge curves are checked and recalibrated during site visits several times per year."

RC: Eq. (2) I am not familiar with the 'inf' notation. Please explain
AR: inf stands for infimum, also known as the greatest lower bound in a set. We have added this to the variable explanation for equation 2 (page 5, ll 30-31).

RC: Page 9, line 16: Please explain what ' a rough indication of the left tail properties' really means.
AR: This is explained in more detail in section 2.7. We have removed the term "rough" and added a reference to section 2.7 (page 7, l 22), where it is explicitly indicated what properties of the left tail can be inferred from the minimum observable scale.

RC: Page 11, line 12: A method from can 2004 hardly be referred to as a classic
AR: We agree that it is hard to define a historical reference as to when a method can be termed classic. We replaced the term "classical" by "frequently used" (page 9, l2).

RC: Page 12, line 8: 'of the log-scale of ln(\lambda):'
AR: thanks for pointing this out, we have added this.

RC: Figure 2: I think the two graphs looks quite similar, i.e. full of black lines. Maybe chose, say, one year to better bring out the differences
AR: We have reduced the time period for these graphs to 5 years for better visualisation. We preferred to use 5 years instead of one year, to visualise the effect of seasonal variation and wet versus dry years.

RC: Section 3.5: In general, I think this section needs more grounding in previous studies to better emphasise which findings are novel and which once are already well-known and even expected. Every hydrological model I know scales flow magnitude with basin area - anything else would be very surprising. There are several studies linking flow variation to basin areas as well, notably CV of flood events (e.g. Blöschl and Sivapalan, 1997). Basin control on skewness is generally more difficult to identify due to sampling variability of skewness indicators. With regards to urbanization, most models assume increasing urbanisation result in increased runoff. Notably Kjeldsen (2010) reported that this is not always the case, and sometime flood magnitude is less than expected in an urban basin. Kjeldsen (2010) also investigated the effect of urbanization on higher order moments (CV and skewness) and also reported a decrease in skewness for more urban basins.
AR: This comment seems to refer mainly to sections 3.2, where we discussed CV and skewness indicators for IATs and flows rather than section 3.5 where flashiness indicators are compared. The reviewer has a valid point that results should be related to findings in previous studies, thanks for pointing out a couple of relevant references. We have added this text to section 3.2, relating the results we found to previous findings in the literature (page 12-13, ll 31-10):
"This confirms results previously reported in the literature on scaling between flows and basin area

(e.g. Goodrich et al. (1997), Smith (1992) and specifically between CV of flows and basin area (Bloschl and Sivapalan (1997). These authors also refer to complexities in hydrological response resulting in deviations from this general relationship. The same applies for the basins in our study, where basin area only explains part of the flow variability, especially for smaller basins."

"While Kjeldsen (2010) reported a decrease in CV and skewness associated with urbanization for basins in the UK, we did not find significant correlations based on CV and skewness indicators for flows. Skewness for IATs was significantly negatively correlated with imperviousness; as stated before, this is probably associated with IAT statistics being more sensitive to variability in high flows than conventional flow statistics."

RC: Page 24, 2. paragraph: The attribution of results to specific features of the urban environment (dams, detension ponds, etc) is speculative as this has not been investigated in the study
AR: We have removed this interpretation from conclusion nr.2, as it is more elaborately discussed in conclusion nr. 3. We have rephrased this point in conclusion nr 3 as follows (page 20, ll 16-19):
"Negative correlation between CV values of IATs and imperviousness probably indicates stronger influence of flow regulation by detention and capacity constraints of stormwater drains in more urbanised basins, resulting in more uniform runoff during rainy periods. IATs during these periods concentrate relatively more closely to the mean and show fewer extremes."

RC: Page 24, line 26-27: It would be more convincing if 'contrary to finding of many studies' could be made more factual with references to which findings and what studies.
AR: The reviewer is right that this point needs more elaboration. We have added a brief discussion of findings in the literature, including references that found results similar to ours as well as others that reached different conclusions (page 20, ll 19-26):
"This result is contrary to findings reported in the literature, where urbanisation tends to be associated with higher peak flows (e.g. Rose, 2001, Cheng, 2002, Du, 2012, Huang 2008). On the other hand, several studies have found mixed effects of urbanisation on flow peaks associated with a combination of imperviousness and flood mitigation measures, especially for basins in the US where urbanisation has predominantly taken place after implementation of stormwater legislation to lower peak discharges (e.g Smith, 2013, Hopkins, 2015, Miller, 2014). For the basins in Charlotte watershed, urbanisation has taken place before as well as after stormwater legislation and a combination of flow regulation by detention facilities and peak flow restrictions induced by capacity constraints results in an overall effect of peak flow reduction associated with urbanisation."

**Statistical analysis of hydrological response in urbanising catchments based on adaptive sampling using inter-amount times**

Marie-claire ten Veldhuis[1,3] and Marc Schleiss[2,3]

[1]Delft University of Technology, Watermanagement Department
[2]Delft University of Technology, Geosciences and Remote Sensing Department
[3]Princeton University, Hydrometeorology Group

*Correspondence to:* Marie-claire ten Veldhuis (j.a.e.tenveldhuis@tudelft.nl)

**Abstract.** Urban catchments are typically characterised by a more flashy nature of the hydrological response compared to natural catchments. Predicting flow changes associated with urbanisation is not straightforward, as they are influenced by interactions between impervious cover, basin size, drainage connectivity and stormwater management infrastructure. In this study, we present an alternative approach to statistical analysis of hydrological response variability and basin flashiness, based on the distribution of inter-amount times. We analyse inter-amount time distributions of high-resolution streamflow time series for 17 (semi)urbanised basins in North Carolina, US, ranging from 13 km$^2$ to 238 km$^2$ in size. We show that in the inter-amount times framework, sampling frequency is tuned to the local variability of the flow pattern, resulting in a different representation and weighting of high and low flow periods in the statistatical distribution. This leads to important differences in the way the distribution quantiles, mean, coefficient of variation and skewness vary across scales and results in lower mean intermittency and improved scaling. Moreover, we show that inter-amount times distributions can be used to detect regulation effects on flow patterns, identify critical sampling scales and characterise flashiness of hydrological response. The possibility to use both the classical approach and the inter-amount time framework to identify minimum observable scales and analyse flow data opens up interesting areas for future research.

**1 Introduction**

Hydrological response in urban catchments tends to be more flashy compared to natural ones as a result of their higher degree of imperviousness. Increase in flashiness is typically characterised by shorter response times to rainfall, higher runoff ratios and higher peak flows (Berne et al., 2004; Smith et al., 2005). On the other hand, high impervious degrees may reduce base flows and lead to intermittent flow during dry periods. At the same time, urbanisation is usually tied to development of urban drainage infrastructure, associated with artificial flow control as well as higher peak flows due to increased drainage connectivity. Predicting the degree of flashiness or base flow reduction associated with urbanisation is not straightforward, as it depends on the interplay of impervious cover, basin size and shape, soil properties, basin slope, drainage connectivity and control structures such as detention ponds, weirs and pumps (Emmanuel et al., 2012; Fletcher et al., 2013; Smith et al., 2013). Traditional analyses of flow time series tend to focus on specific aspects and flow characteristics, aiming for example at predicting low flow durations or peak flow magnitudes. For analysis of change in hydrological response, it may be beneficial to combine both

peak flow and low flow statistics into a single framework. This applies in particular to the context of urban hydrology where urbanisation and human intervention alter both high flow and low flow characteristics of the hydrological response. Combining both aspects in a single analysis is difficult, as flow distributions are highly skewed and frequencies of low and high flow values are very different. In this paper, we show how alternative sampling of flow time series based on inter-amount times leads to

5   more balanced statistical distributions, better representation of both high and low flows in a single framework and more robust behaviour of statistical distributions across scales.

**1.1   Statistical analysis of hydrological response**

Many authors have investigated methods for characterising hydrological response and changes therein, including univariate analysis and multivariate statistics, combining several hydrograph properties such as flood peak, flood volume and flood dura-

10  tion (e.g., Salvadori and De Michele, 2004; Favre et al., 2004; Grimaldi and Serinaldi, 2006; Vittal et al., 2015). Traditional statistical analysis techniques tend to focus on either left or right tail properties of statistical distributions, but not necessarily using the same statistical framework. Low flow analyses for example are primarily concerned with the total time the flow stays below a critical threshold (see e.g. Smakhtin (2001) for an extensive review). By contrast, peak flow analysis puts more weight on total accumulated flows at a given time scale using annual flow maxima or peak-over-threshold values to derive extreme

15  value statistics and establish flood frequency curves (e.g., Stedinger, 1983; Lang et al., 1999; Villarini et al., 2009; Smith and Smith, 2015). Both approaches are valid and solidly rooted in the context of extreme event analysis with numerous applications in drought and flood risk analysis. However, the statistical frameworks they rely on are not necessarily the same. Low flow analysis favours 'time' as a random variable. Peak flow analysis on the other hand treats the 'flow amount' over a fixed time interval as the main random quantity. This might seem more intuitive to many but there is no strong compelling reason to prefer

20  one approach over the other a priori. For example, one might as well adopt an alternative framework in which the unknown random variable is the 'time' necessary to cumulate a fixed, critical amount of flow. [c1]By doing so, both low flows and peak flows can be analysed using the same statistical framework. This approach is known as the inter-amount time (IAT) method (Schleiss and Smith, 2016) and has been previously proposed to analyse the properties of intermittent rainfall time series. An important goal of this paper is to derive properties of statistical distributions obtained by applying the IAT formalism to flow

[revised manuscript text omitted]
. [c1]These curves were established based on protocols developed by USGS and include manual flow measurements during site visits performed by USGS staff. As part of this procedure, stage-discharge curves are checked and recalibrated during site visits several times per year (https://waterdata.usgs.gov/nwis/measurements). The percentage of missing flow data was smaller than 5% for all gauges included in the analysis; missing data were treated like zeros. The effect of missing data on IATs is difficult to predict as this depends on the pattern of missing values and whether or not they occur during a period of low or peak flow. Sensitivity studies by Schleiss and Smith (2016) have shown that the general effect of replacing missing values by zeros is that a few sample IATs will be overestimated. This mostly affects the right tail of the distribution and tends to have limited impact on peak flow characteristics. Another strategy would be to replace missing values by mean or median flow value, which may slightly reduce the overestimation of IATs in case several missing values occur in row. However, in this paper only the worst case scenario will be considered, i.e. missing values were replaced by zeros.

**2.2 Definition of inter-amount times**

In this paper we analyse hydrological flow variability, based on the distribution of inter-amount times. We use the following definition of inter-amount time (IATs), based on Schleiss and Smith (2016): Let $\Delta q > 0$ denote a fixed flow amount. We define the series of IATs $\tau_n(\Delta q)$ with respect to $\Delta q$ as follows:

$$\tau_n(\Delta q) = t_n(\Delta q) - t_{n-1}(\Delta q) \tag{1}$$

[revised manuscript text omitted]

It is worth pointing out that the lower bound on the inter-amount in (9) also provides an [c1] indication of the left-tail properties of IATs, thus of the degree of flashiness of the hydrological response, i.e. the smallest scale at which flow variations can be studied given a fixed temporal observational resolution. [c2]We will elaborate on this in section 2.7, [c3]where we discuss this property in relation to basin flashiness. 
[revised manuscript text omitted]
. [c1]This confirms results previously reported in the literature on scaling between flows and basin area (e.g., Goodrich et al., 1997; Smith, 1992) [c2]and specifically between CV of flows and basin area (Bloeschl and Sivapalan (1997)). [c3]These authors also refer to complexities in hydrological response resulting in deviations from this general relationship. The same applies for the basins in our study, where basin area only explains part of the flow variability, especially for smaller basins. Results showed that larger imperviousness is associated with higher mean flows and significantly lower CV-values for IATs, [c4]while there was not significant correlation between CV-values for flows and imperviousness. This is probably explained by urbanisation being mainly associated with stronger flow regulation by [c5] detention [c6][c7]and capacity constraints in the drainage system. Since IATs are relatively more sensitive to high flows, this effect showed up more clearly in CV values for IATs than for flows. CV and skewness values are much higher for flows than for IATs, while medcouple values are lower for flows, indicating strong asymmetry of the flow distributions and low representation of high flow extremes in the statistical distribution. [c8]While (Kjeldsen, 2010) [c9]reported a decrease in CV and skewness associated with urbanization for basins in the UK, we did not find significant correlations based on CV and skewness indicators for flows. Skewness for IATs was significantly negatively correlated with imperviousness; as stated before, this is probably associated with IAT statistics being more sensitive to variability in high flows than conventional flow statistics.

**3.3   Distribution of changes in inter-amount times**

Figure 6 shows histograms of first-order differences in IATs and flows at the 24 hour analysis scale, for Irvins Creek, the least urbanised basin, LSugarM the most impervious basin, Stewart Creek, a basin with low flow regulation and McAlpine, the largest of all studied basins. In the flow histograms, negative differences are associated with recession, positive differences with flow rise. Conversely, negative differences in IATs occur during flow rise, positive differences during flow recession. Most flow differences are concentrated in the 0 to -0.5 mm bin, associated with slow flow recession of 0.5 mm/day. Most IAT differences are concentrated in the 0 to 0.1 or 0.2 day bin, associated with steeper flow recession of approximately 5 to 10 mm per day. This
* * *
[c1] *Text added.*
[c2] *Text added.*
[c3] *Text added.*
[c4] *Text added.*
[c5]
[c6]
[c7] *Text added.*
[c8] *Text added.*
[c9] *Text added.*

[revised manuscript text omitted]

[c2]

[c3]

3. Comparison of coeffients of variation (CV) across the 17 basins showed that CV values of flows were significantly negatively correlated with basin size. CV values of IAT distributions were not significantly correlated with basin size. This was explained by basin size having a stronger smoothing effect on low flow variability, strongly represented in conventional flow time series, than on peak flows that are more frequently represented in IAT time series. By contrast, CV values of IAT distributions were negatively correlated with imperviousness, while correlation between CV values for flows and imperviousness was not significant. Negative correlation between CV values of IATs and imperviousness probably indicates stronger influence of flow regulation [c4]by detention and capacity constraints of stormwater drains in more urbanised basins, resulting in more uniform runoff during rainy periods. IATs during these periods concentrate relatively more closely to the mean and show fewer extremes. [c5]This result is contrary to findings reported in the literature, where urbanisation tends to be associated with higher peak flows (e.g., Rose and Peters, 2001; Cheng and Wang, 2002; Du et al., 2012; Huang et al., 2008). [c6]On the other hand, several studies have found mixed effects of urbanisation on flow peaks associated with a combination of imperviousness and flood mitigation measures, especially for basins in the US where urbanisation has predominantly taken place after implementation of stormwater legislation to lower peak discharges (e.g., Smith et al., 2013; Hopkins et al., 2015; Miller et al., 2014). [c7]For the basins in Charlotte watershed, urbanisation has taken place before as well as after stormwater legislation and a combination of flow regulation by detention facilities and peak flow restrictions induced by capacity constraints results in an overall effect of peak flow reduction 
[revised manuscript text omitted]

Bloeschl, G., Sivapalan, M., 1997. Process controls on regional flood frequency: Coefficient of variation and basin scale. Water Resources Research 33 (12), 2967–2980, cited By 78.

Brys, G., Hubert, M., Struyf, A., 2004. A robust measure of skewness. Journal of Computational and Graphical Statistics 13 (4), 996–1017.

Cheng, S.-J., Wang, R.-Y., 2002. An approach for evaluating the hydrological effects of urbanization and its application. Hydrological Processes 16 (7), 1403–1418, cited By 56.

Davies, R., 2002. Hypothesis testing when a nuisance parameter is present only under the alternative: Linear model case. Biometrika 89 (2), 484–489.

Dippe, M. A., Wold, E. H., 1985. Antialiasing through stochastic sampling. Computer Graphics (ACM) 19 (3), 69–78.

Du, J., Qian, L., Rui, H., Zuo, T., Zheng, D., Xu, Y., Xu, C.-Y., 2012. Assessing the effects of urbanization on annual runoff and flood events using an integrated hydrological modeling system for qinhuai river basin, china. Journal of Hydrology 464-465, 127–139, cited By 63.

Emmanuel, I., Andrieu, H., Leblois, E., Flahaut, B., 2012. Temporal and spatial variability of rainfall at the urban hydrological scale. Journal of Hydrology 430-431, 162–172.

Favre, A.-C., Adlouni, S., Perreault, L., Thiémonge, N., Bobée, B., 2004. Multivariate hydrological frequency analysis using copulas. Water Resources Research 40 (1).

Feizi, S., Angelopoulos, G., Goyal, V., Medard, M., 2011. Energy-efficient time-stampless adaptive nonuniform sampling. pp. 912–915.

Fletcher, T., Andrieu, H., Hamel, P., 2013. Understanding, management and modelling of urban hydrology and its consequences for receiving waters: A state of the art. Advances in Water Resources 51, 261–279.

Goodrich, D., Lane, L., Shillito, R., Miller, S., Syed, K., Woolhiser, D., 1997. Linearity of basin response as a function of scale in a semiarid watershed. Water Resources Research 33 (12), 2951–2965, cited By 149.

Grimaldi, S., Serinaldi, F., 2006. Asymmetric copula in multivariate flood frequency analysis. Advances in Water Resources 29 (8), 1155–1167.

Gustafson, D., Carr, K., Green, T., Gustin, C., Jones, R., Richards, R., 2004. Fractal-based scaling and scale-invariant dispersion of peak concentrations of crop protection chemicals in rivers. Environmental Science and Technology 38 (11), 2995–3003.

Hopkins, K., Morse, N., Bain, D., Bettez, N., Grimm, N., Morse, J., Palta, M., Shuster, W., Bratt, A., Suchy, A., 2015. Assessment of regional variation in streamflow responses to urbanization and the persistence of physiography. Environmental Science and Technology 49 (5), 2724–2732.

Huang, H.-J., Cheng, S.-J., Wen, J.-C., Lee, J.-H., 2008. Effect of growing watershed imperviousness on hydrograph parameters and peak discharge. Hydrological Processes 22 (13), 2075–2085, cited By 37.

Kjeldsen, T., 2010. Modelling the impact of urbanization on flood frequency relationships in the uk. Hydrology Research 41 (5), 391–405, cited By 16.

Labat, D., Hoang, C., Masbou, J., Mangin, A., Tchiguirinskaia, I., Lovejoy, S., Schertzer, D., 2013. Multifractal behaviour of long-term karstic discharge fluctuations. Hydrological Processes 27 (25), 3708–3717.

Labat, D., Mangin, A., Ababou, R., 2002. Rainfall-runoff relations for karstic springs: Multifractal analyses. Journal of Hydrology 256 (3-4), 176–195.

Lang, M., Ouarda, T., Bobée, B., 1999. Towards operational guidelines for over-threshold modeling. Journal of Hydrology 225 (3-4), 103–117.

5 Lombardo, F., Volpi, E., Koutsoyiannis, D., Papalexiou, S., 2014. Just two moments! a cautionary note against use of high-order moments in multifractal models in hydrology. Hydrology and Earth System Sciences 18 (1), 243–255.

Miller, J., Kim, H., Kjeldsen, T., Packman, J., Grebby, S., Dearden, R., 2014. Assessing the impact of urbanization on storm runoff in a peri-urban catchment using historical change in impervious cover. Journal of Hydrology 515, 59–70.

Pandey, G., Lovejoy, S., Schertzer, D., 1998. Multifractal analysis of daily river flows including extremes for basins of five to two million square kilometres, one day to 75 years. Journal of Hydrology 208 (1-2 /2), 62–81.

Poff, N., 2002. Ecological response to and management of increased flooding caused by climate change. Philosophical Transactions of the Royal Society A: Mathematical, Physical and Engineering Sciences 360 (1796), 1497–1510.

Praskievicz, S., Chang, H., 2009. A review of hydrological modelling of basin-scale climate change and urban development impacts. Progress in Physical Geography 33 (5), 650–671.

15 Richter, B., 1996. A method for assessing hydrologic alteration within ecosystems [un metro para evaluar alteraciones hidrologicas dentro de ecosistemas]. Conservation Biology 10 (4), 1163–1174.

Rose, S., Peters, N., 2001. Effects of urbanization on streamflow in the atlanta area (georgia, usa): A comparative hydrological approach. Hydrological Processes 15 (8), 1441–1457, cited By 199.

[revised manuscript text omitted]